# 1q gain bypasses the selective barrier against aneuploidy in RPE differentiation via wild-type co-culture rescue

Edouard Couvreu de Deckersberg[1,6], Yingnan Lei[1,6], Nuša Krivec[1], Anfien Huyghebaert[1], Mai Chi Duong [1,2], Charlotte Janssens [1], Marius Regin [1], Olga Tsuiko[3], Kiavash Movahedi [4], Stefaan Verhulst [5], Leo A. van Grunsven [5], Karen Sermon[1], Diana Al Delbany[1] & Claudia Spits [1] ✉

Human pluripotent stem cells (hPSC) are increasingly used in clinical trials, with retinal pigment epithelium (RPE) being among the most transplanted cell types. However, hPSC frequently acquire chromosomal abnormalities, whose impact on differentiation and transplant safety is incompletely understood. In this work, we investigate how aneuploidy influences the progression of hPSC through undirected RPE differentiation. Large-scale omics analysis of genetically normal hPSC cultures reveals pervasive low-grade mosaicism, with 3–6% of cells carrying different aneuploidies. During undirected differentiation, all aneuploid cells are lost except those with a gain of chromosome arm 1q. These cells differentiate efficiently only when co-cultured with wild-type cells, which promote neural differentiation through paracrine signals that 1q-gain cells are uniquely receptive to. This interaction enables 1q-gain cells not only to persist but to eventually dominate the culture, owing to their competitive advantage.

Over a hundred clinical trials are ongoing or completed involving the transplantation of cells derived from human pluripotent stem cells (hPSC), all of which are phase I/II trials focused on safety[1]. Currently, most involve the use of retinal pigmented epithelium (RPE) cells derived from hPSC to restore or improve vision in patients suffering from retinal degenerative diseases, including age-related macular degeneration and Stargardt's macular dystrophy[1].

A remaining safety concern of these treatments is the susceptibility of hPSC to accumulate genomic abnormalities. The most common abnormalities involve segmental or full gains in chromosomes 1, 12, 17, and 20, but also include point mutations and epigenetic changes, often resembling mutations found in cancers[2]. Despite their recurrence, not much is known about their downstream functional consequences, obscuring their impact on both the research and the clinical applications of hPSC[2]. Only two of these recurrent

abnormalities have been thoroughly characterized in the undifferentiated state. Gain of chromosome 12 results in increased hPSC proliferation rates and altered transcriptomic profiles, likely due to *NANOG* over-expression[3]. Higher levels of Bcl-xL in hPSCs with a gain of 20q11.21 result in decreased sensitivity to apoptosis[4], which confers a survival advantage to the cells[5]. Regarding their impact on differentiation, hPSCs with a gain of 20q11.21 have impaired neuroectoderm commitment without affecting mesendoderm induction[6,7], and it has been shown that cells with an isochromosome 20q (iso20q) fail to survive RPE differentiation[8]. Even when chromosomally abnormal hPSC are capable of differentiating, they display altered gene-expression patterns suggestive of malignant transformation[2]. Our current understanding of how these mutations impact the oncogenic potential of hPSC-derived cells remains limited. Research on this topic has predominantly focused on tumor formation by either residual

[1]Research Group Reproduction, Genetics and Development, Faculty of Medicine and Pharmacy, Vrije Universiteit Brussel, Brussels, Belgium. [2]Department of Biochemistry, Military Hospital 175, 786 Nguyen Kiem Street, Ho Chi Minh City, Vietnam. [3]Reproductive Genetics Unit, Centre of Human Genetics, University Hospital Leuven, Leuven, Belgium. [4]Brain and Systems Immunology Lab, Brussels Center for Immunology, Faculty of Medicine and Pharmacy, Vrije Universiteit Brussel, Brussels, Belgium. [5]Liver Cell Biology Research Group, Faculty of Medicine and Pharmacy, Vrije Universiteit Brussel, Brussels, Belgium. [6]These authors contributed equally: Edouard Couvreu de Deckersberg, Yingnan Lei. ✉e-mail: claudia.spits@vub.be

undifferentiated cells or highly proliferative progenitor cells in the differentiated cell product[9]. Conversely, the abnormalities seen in hPSCs could be regarded as a first hit in cancerous transformation. Transplanted cells with genetic abnormalities could be more likely to undergo oncogenic transformation, requiring fewer genetic hits to initiate the process[2,9]. In line with this, recent work in mice has shown that aneuploidy drives teratoma metastasis, with multiple organ dissemination[10].

Currently, all clinical centers subject their hPSC cultures and hPSC-based products to genetic screening prior to use in patients. This typically involves the use of G-banding and, more recently, massively parallel sequencing (MPS), which enables the detection of copy number variation (CNV) as well as potentially harmful single-nucleotide changes. Incidentally, the first clinical trial using hPSC-derived RPE was halted after potentially harmful mutations were identified in both the hPSC and the RPE cells derived from them[11]. Standard methods for genetic screening cannot detect low-grade mosaic abnormalities in hPSC cultures, which are common[12–15]. Since specific abnormalities may lead to poor differentiation or increased malignant potential, their presence in a low-grade mosaic could result in the transplantation of a cell product that, at best, has decreased effectiveness, and at worst, has tumor-initiating properties.

In this study, we hypothesized that while some CNVs render hPSC refractory to correct differentiation, others may confer a growth advantage during cell specification, leading to the enrichment of specific mutant hPSC during differentiation. We investigated these events during RPE differentiation to assess how genetic makeup influences progression to a clinically relevant cell type.

## Results

### hESC yield neural cells with line-specific signatures

We differentiated genetically balanced hESC into RPE using an unguided differentiation protocol[16] (Supplementary Table 1 and Supplementary Fig. 1). After 4–8 weeks, pigmented areas indicated clusters of cells that differentiated spontaneously into RPE. These clusters were picked and passaged to yield a pure RPE cell culture (Fig. 1a, b). We wanted to establish if these clusters are of clonal origin, as this would generate a genetic bottleneck whose size depends on the number of clones picked for the final RPE culture. For this, we differentiated an hESC line labeled with the RGB (red-green-blue) lineage tracing system[17]. After eight weeks, the pigmented colonies were predominantly of clonal origin, as indicated by same color within the cluster (Fig. 1c). Single-cell RNA sequencing (scRNAseq) of two genetically balanced cultures at this stage showed besides RPE and retinal progenitor cells, other neural cell types, most closely resembling cortical hem, oligodendrocyte progenitors, amacrine and retinal ganglion cells, as well as a population of amnion-like cells (Fig. 1d, e). These cell types are excluded from the final RPE product by the colony picking and passaging. After two passages, the cultures consist of homogeneous layers of RPE cells, characterized by their typical cobblestone morphology (Fig. 1f).

We next differentiated five genetically balanced hESC lines to RPE and stained the purified cultures for markers of neuroectoderm (PAX6), RPE (PMEL, ZO-1, and BEST1) and pluripotency (NANOG) (example shown in Fig. 1g, all lines in Supplementary Fig. 2). All RPE cultures were fully negative for NANOG, while ZO-1 and PMEL were expressed by all cells. All cultures presented two cell populations, PAX6$^{high}$/BEST1$^{low}$ and PAX6$^{low}$/BEST1$^{high}$, reflecting different levels of maturation, as seen in other studies[18]. We performed scRNAseq for these five RPE cultures, which yielded high-quality transcriptomic profiles for 87,168 cells. Additionally, previously generated scRNA-seq data from one hESC line ($N = 1996$) served as a reference for the undifferentiated state. UMAP of all RPE cells showed each line predominantly clustered separately, with four small clusters originating from multiple cell lines (Fig. 1h). We first compared RPE to

undifferentiated hESC to confirm the induction of RPE-related genes and loss of pluripotency (Supplementary Fig. 3a). We found that the five RPE cultures expressed genes associated with key RPE functions, such as pigment synthesis, visual cycle, phagocytosis, ion channel and tight junctions[19] (Fig. 1i, Supplementary Table 2). We next integrated our dataset into that of Senabouth et al., which includes scRNAseq of RPE derived from 79 different iPSC lines[18] (Fig. 1j). The results show that our cells express RPE markers at similar levels to those reported in this large study. In addition, and consistent with the PAX6/BEST1 immunostaining (Fig. 1g), the UMAPs show the presence of two distinct populations of RPE based on maturation state in both datasets. Importantly, no RPE cells showed residual pluripotency gene expression (Fig. 1j, k). Taken together, the cell morphology, immunostaining results and gene-expression profiling indicate that our cells differentiated efficiently into RPE.

We analyzed gene expression differences between clusters of RPE cells from different hESC lines. We compared each cluster to the other four and identified 1206 genes that drove the differences between lines. The Venn diagram in Fig. 1k shows that the RPE derived from each hESC line expresses a specific set of genes, with few overlaps and each line has between 12 and 32% unique deregulated genes. The heatmap of differentially expressed genes in Fig. 1l illustrates that each cell line has its own transcriptomic signature. Gene set enrichment analysis revealed no enrichments for pathways related to RPE functionality or stem cell differentiation (Supplementary Table 3). Removing line-specific genes before UMAP plotting reduced separation between lines, while the small clusters composed of cells from different cell lines remained apparent (Fig. 1m). These differences are likely of minor biological significance and may be due to genetic and epigenetic variation between cell lines[20].

Next, we identified the four clusters composed of cells from different lines and that had transcriptomic signatures that differed from RPE cells (Fig. 1n). Cluster 2 contains cells originating from all 5 hESC cell lines and is the only one showing high expression of *MKI67*, and cell cycle analysis showed that 47% of these cells were in G2/M and 53% in S phase, indicating that they are proliferating cells (Fig. 1o, Supplementary Fig. 3b, c). We assigned each cluster a score related to different neural and ocular cell types[21] (Supplementary Table 4). All RPE clusters, as well as cluster 2, show the highest scores for the RPE gene set. Clusters 1 and 3 show high expression of astrocyte markers, and cluster 3 also scored highly for Schwann and Mueller glia cells. Finally, cluster 4 expressed genes marking pigmented ciliary body cells, a cell type related to the RPE lineage (Fig. 1p, Supplementary Fig. 3d). Together, these findings show that while most cells in the purified cultures are RPE, a subset displays transcriptional features indicative of other neural lineages.

### RPE differentiation eliminates aneuploid cells except for 1q gains

We next investigated the genetic composition of the cell cultures at the single-cell level before and after RPE differentiation. We performed single-cell DNA sequencing (scDNAseq) on three of the five hESC lines at the start of the differentiation process. We were unable to perform scDNAseq on the two remaining lines and on the differentiated cells due to the discontinuation of the essential 10× Genomics products required for this specific analysis. Therefore, we employed the inferCNV algorithm on scRNAseq data from RPE cells derived from all five hESC lines to assess the genetic content at the endpoint of differentiation.

The analysis yielded high-quality scDNAseq results for 1668 cells; their karyotypes are illustrated in Fig. 2a–c. The cells could broadly be categorized as cells with genetically balanced content, cells with chaotic genetic content (multiple abnormalities spanning more than 10% of the genome), and cells with other gains or losses. We subdivided the abnormal non-chaotic cells by whether the CNVs were

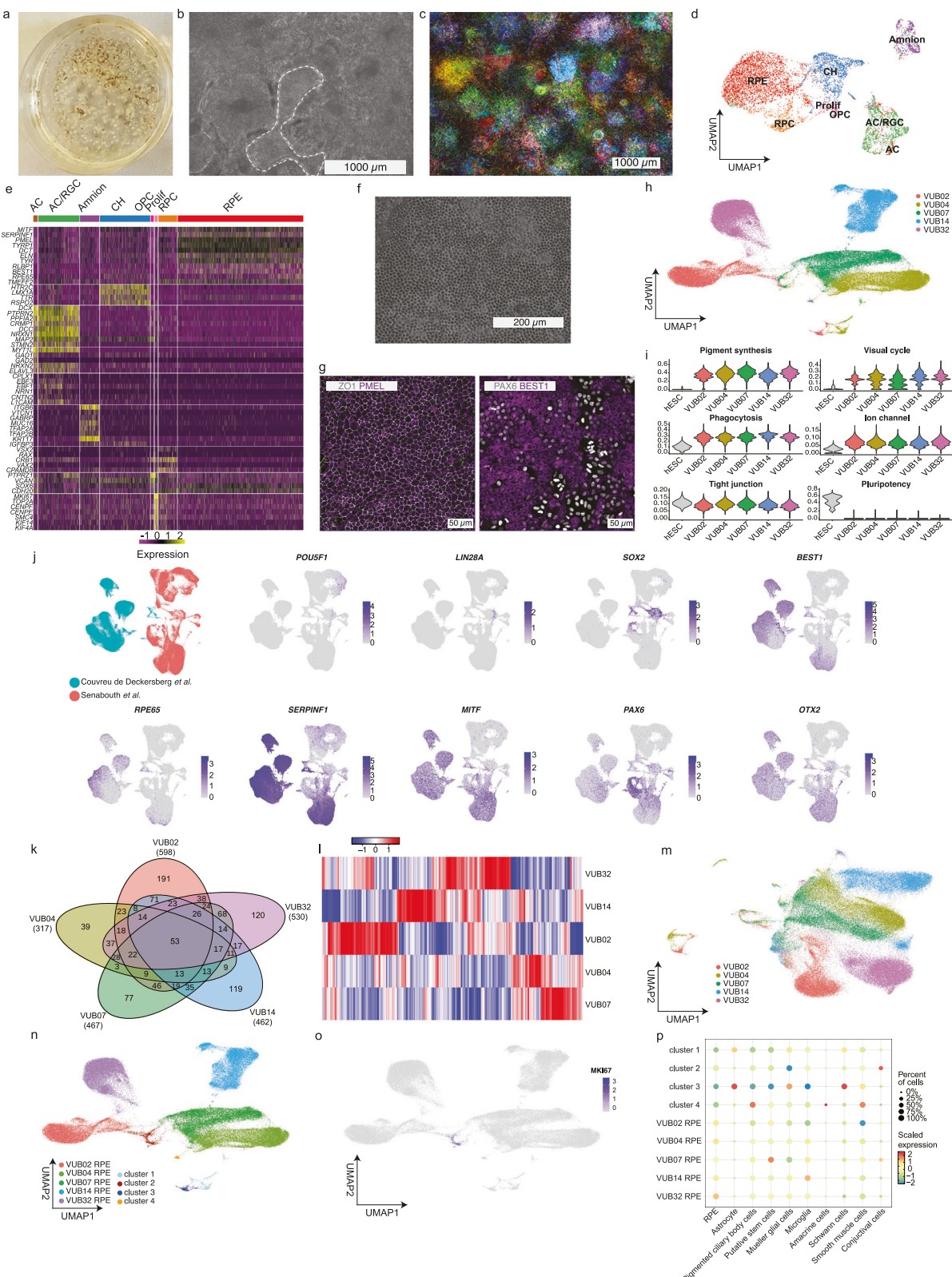

larger than 20 Mb and thus detectable by inferCNV. This facilitated comparisons between the outcomes of these two distinct analyses. Figure 2d provides an example of diverse cell types, including chaotic cells, cells with gains in chromosome 1q, cells with iso20q, and cells with segmental gains in chromosomes 15q and 20q.

On average, 95% of cells were genetically balanced, with minor variations observed between the three hESC lines (Fig. 2e).

Interestingly, cells with chaotic genetic content were found in all three lines. These chaotic cells were similar to those found in human pre-implantation embryos, where they originate from abnormal mitotic events with multipolar spindles[22]. To investigate whether similar abnormal cell divisions occurred in hESC, we conducted live-cell imaging for 24 h to visualize DNA and microtubules (Supplementary Movie 1). Figure 2f shows a cell undergoing tripolar spindle mitosis,

**Fig. 1 | Genetically balanced hESC lines differentiate into RPE with individual gene-expression signatures as well as astrocytes and pigmented ciliary body cells. a** Example of a 9 cm² culture dish, 8 weeks into RPE differentiation. **b** Magnification of the cells shown in (**a**), the dotted line marks three adjoining clusters of pigmented cells, which were picked and passaged to establish a pure RPE population. **c** Fluorescence imaging of an hESC line labeled with the RGB lineage tracing system, 8 weeks into RPE differentiation. **d** UMAP of the scRNA sequencing of hESC[wt] after 60 days of spontaneous differentiation, prior to colony picking, containing retinal pigment epithelium cells (RPE), cortical hem cells (CH), amacrine cells (AC), retinal ganglion cells (RGC), amnion-like cells (Amnion), retinal progenitor cells (RPC), oligodendrocyte progenitor cells (OPC) and proliferating cells (Prolif). **e** Heatmap for the expression of marker genes for the cell types identified in panel (**d**). **f** Brightfield imaging of a purified cell culture showing the

RPE cobblestone morphology. **g** Example of the immunostaining of the purified RPE cell cultures for PMEL, ZO-1, PAX6, and BEST1. **h** UMAP of the scRNA sequencing of hESC-derived RPE, labelled by cell line. **i** Violin plots of the expression per cell and per cluster of undifferentiated hESC and RPE marker gene sets. **j** Aggregation of our single-cell RNA sequencing dataset to that of Senabout et al.[18], including 127659 RPE cells derived from 79 induced pluripotent stem cell lines. **k** Venn diagram of the differentially expressed genes across the different hESC-derived RPE. **l** Heatmap of the differentially expressed genes shown in panel (**k**). **m** UMAP of the hESC-derived RPE without the cell-line-specific transcriptomic profiles. **n** UMAP indicating the RPE clusters, and the 4 clusters identified by Seurat and composed of cells from different lines. **o** Cluster 2 shows a higher expression of MKI67. **p** UCell scoring for each cluster in (**k**), with scores for neural and ocular-related cell types.

with additional examples in the supplementary movie, confirming that hESC undergo abnormal multipolar divisions, resulting in daughter cells with chaotic genetic content.

The distribution and location of all gains and losses found in the cells (excluding chaotic cells) are shown in Fig. 2g, and their breakpoints are listed in Supplementary Table 5. While single losses were twice as common as single gains, overall, gains outnumbered losses 68 to 40. All lines contained single cells with abnormalities known to confer a growth advantage, potentially leading to culture dominance by cell competition. VUB02, for instance, included cells with a trisomy 12, a trisomy 20, a dup(1)(q32.1q44), and a del(18)(q21.2q22.1). VUB04 had a dup(1)(q21.3q44) and an iso20q and VUB07 had dup(20) (q11.21q13.2) cells.

Remarkably, inferCNV analysis at the endpoint of RPE differentiation revealed that, in contrast to undifferentiated hESC, the only genetic imbalance was different-sized gains of chromosome 1q, found in all cell clusters of VUB04 and VUB07 (Fig. 2h–l). VUB04 exhibited 42.8% of RPE cells with a 77.8 Mb gain of 1q21.3q44 and 2.95% of cells with a 20.4 Mb gain of 1q21.2q24.2. For VUB07, 2.66% of the cells displayed an 82.0 Mb gain of 1q21.3q44. Additionally, the clusters representing cells mis-specified as astrocyte and ciliary body-like cells contained 12.8% of cells with the 77.8 and 82.0 Mb 1q gain and 28.7% of cells with the shorter 20.4 Mb 1q gain. Notably, 1q gain was not enriched in the alternative cell fates, suggesting that this aneuploidy is not the primary driver for appearance (Fig. 2m).

The absence of aneuploidies in RPE other than 1q gain, unlike undifferentiated hESC, was remarkable. To test whether this was due to random colony picking, we calculated the probability of selecting at least one aneuploid clone, assuming equal differentiation potential and no growth advantage. Based on scDNAseq data and ~60 clonal colonies picked per line, these probabilities were 47.8% (VUB02), 96% (VUB04) and 89.7% (VUB07). Given these, the absence of other aneuploidies suggests that not only the differentiation to RPE represents a bottleneck with a strong selective pressure against most aneuploid cells, but that the gain of 1q may be conferring a culture advantage to the cells during this differentiation process.

### RPE with 1q gain shows mild transcriptomic differences

Since a large proportion of RPE cells carried a large gain of 1q (RPE[1q]), we investigated potential transcriptomic differences between RPE[1q] and their wild-type counterparts (RPE[wt]). Cells with a smaller gain of 1q were excluded because of their limited representation in the dataset.

We first assessed the expression of RPE-associated genes and found that RPE[1q] consistently expressed all expected markers (Fig. 3a). The extent of their expression varied, and while all genes except MITF showed statistically significant differences between the two groups, some genes were higher expressed and some lower in RPE[1q] cells. This suggests that these cells differentiated to bona fide RPE, similarly to genetically balanced cells. We conducted single-cell gene set variation

analysis (scGSVA) for the KEGG pathways. Four illustrative examples are shown in Fig. 3b. While many of these pathways reached high statistical significance, the differences in the mean normalized enrichment scores were minimal. We sought to explore whether the observed differences in gene expression were consistently linked to the presence of the 1q gain or if they were the result of cell line variation. For this, we conducted scGSVA analyses on individual cell lines, as well as a comparative analysis of RPE[1q] and RPE[wt] cells in VUB04 and VUB07, as shown in Fig. 3c. Except for the apoptosis gene set, no other gene set consistently diverged, suggesting that the observed variations were primarily attributable to line-to-line variation rather than being directly associated with the presence of the 1q gain. The apoptosis pathway appeared as an exception, with consistent enrichment scores in the isogenic pairs of VUB04 and VUB07 (Fig. 3d). Closer inspection showed that only a subset of genes in the apoptosis pathway were expressed in RPE cells, and these genes did not all consistently differ between RPE[1q] and RPE[wt] cells. Three genes involved in the execution of the apoptotic process, PARP1, MCL1 and LMNA, were significantly upregulated in RPE[1q] cells, and are all located within the region affected by the 1q gain, implying that their increased copy number likely drove higher expression and apoptosis pathway deregulation. Given that these effectors often have opposing roles, predicting their overall biological impact remains challenging.

### Co-culture rescues impaired RPE differentiation of hESC with 1q gain

In the subsequent experiments, we tested whether low-grade mosaic aneuploid hESC observed in genetically normal cultures cannot undergo RPE. For this, we differentiated three wt control lines and nine hESC lines with different recurrent aneuploidies in 44 independent experiments (the list of lines and replicates can be found in Supplementary Table 6).

After 12 weeks, hESC lines carrying a gain of 20q11.21, iso20q, gain of 17q, and derivative chromosomes involving loss of 18q consistently failed to differentiate into RPE (Fig. 4a shows a representative example; additional images in Supplementary Fig. 4). Interestingly, the line with the smallest 1q gain (VUB03[1q32.21]) exhibited pigmented regions characteristic of RPE differentiation, whereas two lines with larger 1q gains (VUB03[1q21.1qter] and VUB01[1q21.1q31.1]) showed no visible pigmentation (Supplementary Fig. 4). At the colony picking stage, we analyzed the mRNA expression levels of RP65, BEST1, PAX6, and POU5F1 and found that genetically balanced lines induced the RPE gene-expression signature, whereas aneuploid cell lines either did not initiate the expression of these genes or did so significantly less than control cells (Fig. 4b). These findings demonstrated that, except for the small focal gain of 1q32.21, aneuploidy impeded the correct differentiation into RPE cells.

Given that earlier experiments identified correctly differentiated RPE harboring large 1q gains when mixed with wt cells, we hypothesized that co-culture with wt would rescue mutant differentiation. We

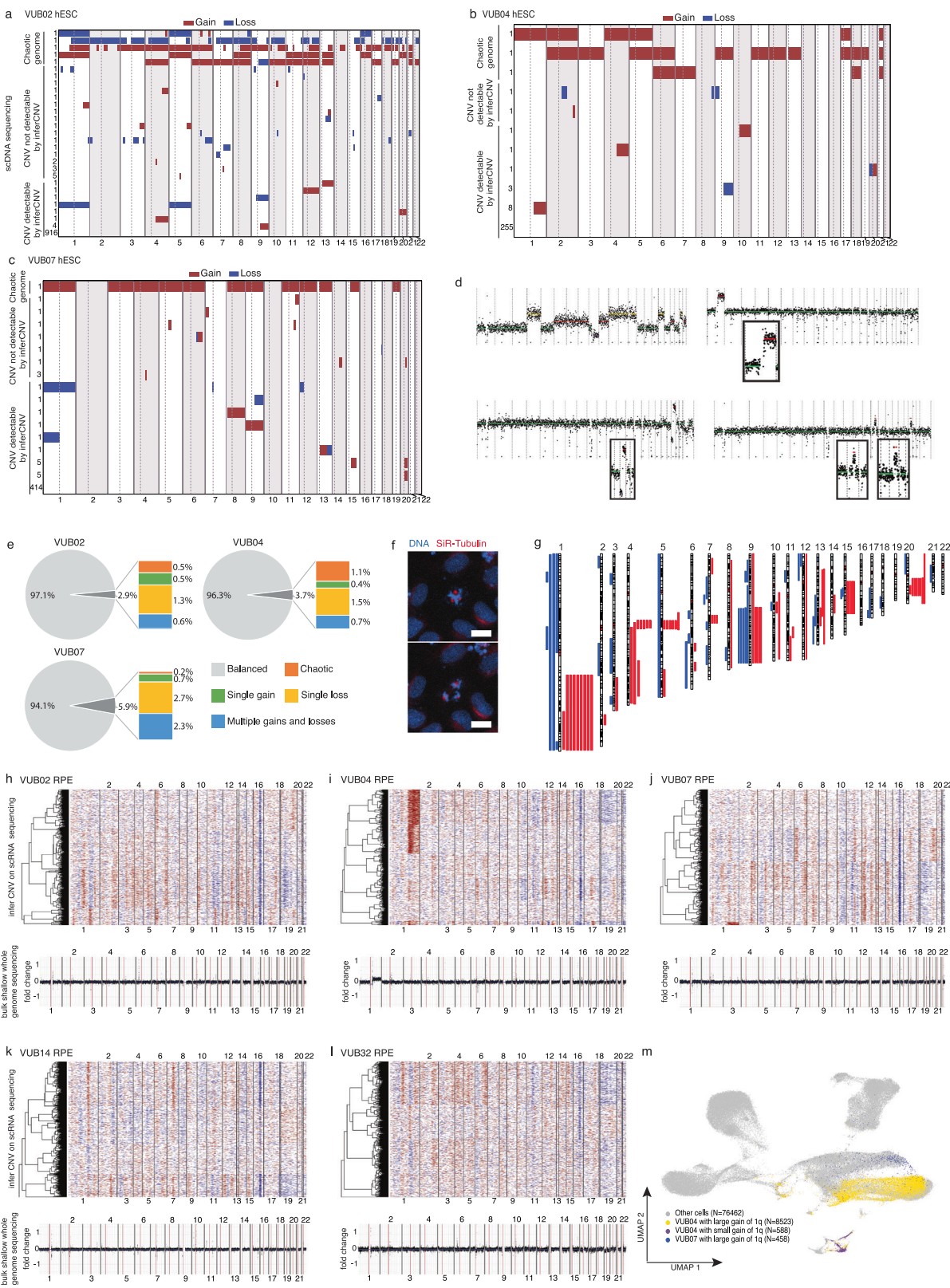

performed scRNA-seq across five time points—day 0, 1, 2, 7, and 14—following spontaneous differentiation. Each day included three conditions: wt alone, 1q-gain alone, and 1q-gain co-cultured with wt at a 1:9 ratio. This was done using two hESC lines, VUB03 and VUB19, for 30 samples, and in total 80686 cells. We confirmed the karyotypes of the cells by inferCNV and found unexpected and substantial genetic mosaicism. VUB03[wt] cultures contained ~5% trisomy 20 cells, VUB19[wt]

cultures were predominantly composed of iso20q cells (82%), with few genetically balanced cells. VUB03[1q] included 62% iso20q and 38% dup1q cells, whereas VUB19[1q] was solely 1q-gain. As a result, at the start of differentiation, the mixed cultures included complex compositions (Fig. 4c). This unintended diversity allowed us to study the early differentiation trajectories of three different types of aneuploid hESC and to track these populations over time in culture.

**Fig. 2 | Genetic variation in the undifferentiated hESC lines and their differentiated RPE cell progeny. a–c** scDNAseq of 956 cells of VUB02, 275 cells of VUB04 and 443 cells of VUB07, prior to differentiation into RPE cells. The plots show the gains (red) and losses (blue) in the autosomes. The number of cells carrying one specific karyotype is shown left of each row, with genetically balanced cells at the bottom row. The karyotypes have been divided into 'chaotic genetic content' (complex karyotype with multiple trisomies and monosomies), 'CNV not detectable by inferCNV' (copy number variations below inferCNV resolution) and 'CNV detectable by inferCNV' (gains and losses above the resolution of inferCNV). **d** scDNAseq results for cells with **a**: chaotic complement, gain of 1q, iso20q and gain at 15q and 20q. **e** Pie charts with percentages of chromosomal abnormalities in the three hESC lines. **f** Still images of live imaging of a tripolar mitosis in hESC (Supplementary Movie 1, scale bar 5 μm). **g** Ideogram of the location of all gains and losses in the hESC lines, excluding chaotic cells. Gains are shown in red and losses in blue. **h–l** inferCNV plots obtained from the scRNAseq of RPE cells derived from VUB02, VUB04, VUB07, VUB14, and VUB32 (top of panel) and their bulk shallow DNA sequencing plot (bottom of panel). **m** UMAP of the scRNAseq of the five RPE cell samples, indicating the genetically different cell populations identified by inferCNV.

The plotting of the relative proportions of the cells over time revealed genetic drifts that differed by karyotype (Fig. 4c). Trisomy 20 and iso20q clones were progressively depleted while the 1q-gain cells outcompeted both wt and other mutant cells, eventually dominating the culture. These results align with earlier findings that spontaneous differentiation imposes a bottleneck that eliminates aneuploid cells such as trisomy 20 and iso20q, while favoring 1q-gain cells.

Next, we analyzed the transcriptional trajectories from the undifferentiated state (day 0) to the first two days of differentiation. UMAP projections of day 0, 1, and 2 samples showed cell clustering primarily driven by karyotype and cell line rather than time point (Fig. 4d, e). VUB03[wt] cells progressed through three distinct clusters corresponding to day 0, 1, and 2 (Fig. 4f). Differential expression analysis across these clusters for genes that consistently up or down-regulated across the days showed only a modest decline in pluripotency-associated gene expression (e.g., *LIN28A*, *POU5F1*). The most prominent transcriptional changes involved genes related to cell adhesion, cytoskeleton, mRNA processing and mTOR signaling, and the upregulation of genes implicated in early neuroectodermal commitment (Supplementary Fig. 5a and Supplementary Table 7). In contrast, all aneuploid cells, including trisomy 20, iso20q cells, and 1q-gain cells grown alone, remained confined to the one cluster, showing minimal or no transcriptional progression from day 1 to day 2 (Fig. 4g–k, overlapping dots obscure visualization). In the VUB03 co-culture, a substantial subset of 1q-gain cells aligned closely with wt cells (Fig. 4l), suggesting that co-culture enabled the rescue of the differentiation trajectory. In the VUB19 co-culture, the 1q-gain cells were co-cultured with iso20q cells, with the resulting population predominantly following the transcriptional trajectory of the iso20q cells (Fig. 4m). These results highlight the profound influence of aneuploidy and cellular context on early differentiation trajectories, with co-culture conditions modulating the fate and transcriptional progression of aneuploid cells.

To gain insight into the regulatory mechanisms underlying this, we performed SCENIC on days 0–2, revealing distinct regulon activity patterns across karyotypes and time points (Fig. 4n, o). Using wt cells as a reference group, including VUB03[wt] and VUB19[wt]. Several regulons were highly active in the undifferentiated state but rapidly lost activity by day 2, including the key regulator of early embryonic development and pluripotency TEAD4[23] and FOSL1, which regulates *SOX2* and *NANOG* expression in cancer cells[24,25]. On the other hand, key neuroectoderm regulators were induced as from day 1. Specifically, NFIB[26] and CUX1[27] showed sustained activity through day 1 before declining, while PBX3[28] and TCF7L1[29] were induced from day 1 to day 2. These regulons were activated in wt cells and in 1q-gain cells co-cultured with VUB03[wt] cells, but not in 1q-gain cells grown alone or in trisomy 20/iso20q cells. In contrast, regulons such as HOXA3 and NKX2-5, associated with mesendodermal fates[30,31], and ZNF599, of unknown function, were preferentially active in iso20q and trisomy 20 cells.

To investigate how altered intercellular signaling contributes to the differentiation of 1q-gain cells co-cultured with wt cells, and the lack of rescue for iso20q and trisomy 20 cells, we analyzed ligand–receptor interactions across days 0–2. We focused on ligands secreted by wt cells and their corresponding receptors in wt, 1q-gain, iso20q, and trisomy 20 cells (Fig. 4p). This analysis revealed a set of ligand–receptor pairs that were active in wt and wt–1q co-cultures but either absent or rewired in iso20q and trisomy 20 cells. For instance, several ligand–receptor pairs involved in neuroectodermal patterning such as NRXN1–NLGN1[32], EFNA5–EPHA6/7[33] and CALM1/2/3-RYR2/CACNA1C[34] were present in wt to wt and wt to 1q-gain communication, whereas iso20q and trisomy 20 cells presented alternative receptor configurations, such as NLGN1–NRXN3, EFNA5–EPHB1, or CALM1/2-INSR, which were driven by different expression levels of the receptors (Fig. 4q–s). These alternative configurations likely reflect differences in signaling specificity and pathway engagement. For instance, NLGN1–NRXN1/3 interactions differ in affinity[35], EPHA and EPHB differ in their signaling pathway activation[36,37], and the insulin receptor (INSR) directly regulates pluripotency and cell commitment[38]. Furthermore, the iso20q and trisomy 20 cells showed specific and alternative ligand–receptor connections, including high levels of FGF2–FGFR2 expression[39], LAMA5[40] and its receptors and PCSK9–LDLR[41], all of which are strongly associated with maintenance of pluripotency. In contrast, 1q-gain cells in monoculture showed patterns of receptor expression closer to wt cells than those of iso20q and trisomy 20 cells, but had lower expression of ligands such as EFNA5 and the calmodulins. Together, these results show that iso20q and trisomy 20 have high expression of pluripotency-maintaining ligands and their receptors, indicating an intrinsic signaling environment favoring self-renewal over differentiation. Moreover, these cells neither express nor appear to respond to neuroectodermal ligands secreted by wt cells, potentially explaining their failure to initiate differentiation under co-culture conditions. In contrast, the rescue of 1q-gain differentiation is mediated by wt-derived ligands that engage compatible receptors on 1q-gain cells, which are absent in 1q-gain cells in monoculture or co-culture with iso20q cells, compromising the initiation of neuroectodermal differentiation.

Last, we studied cell compositions at day 7 and 14 of spontaneous differentiation (Fig. 4t-w, Supplementary Fig. 5b, c). By day 7, four distinct identities emerged: residual undifferentiated, amniotic-like, neuroectoderm and non-neural ectoderm cells. Consistent with the enhanced pluripotency-associated autocrine signaling networks, most of the undifferentiated cells carried trisomy 20, iso20q or were VUB19[1q] cells cultured alone and in co-culture with iso20q cells. By day 14, extraembryonic mesoderm-like cells emerged, exclusively composed of monocultured VUB19[1q]. The neuroectoderm cluster at this stage contained both proliferative and non-proliferative cells, as well as a small cluster of RPE progenitor cells, likely representing the origin of the pigmented colonies observed after two months of differentiation. Most undifferentiated cells at this stage remained confined to the trisomy 20 and iso20q populations. At first sight, the proportion of neuroectodermal cells appeared comparable across different karyotypes at day 14, ranging from 47% to 76% in VUB03[wt], VUB03[1q] (in monoculture or co-culture), and VUB19[wt] or VUB19[1q] in co-culture conditions (Fig. 4x). Conversely, wt cells have the highest neuroectoderm gene expression (Fig. 4z). The neuroectoderm cells derived from VUB03[T20], although representing a small part of the culture, also had a very high average expression of these genes, mostly driven by high levels of SOX2, both a neuroectoderm and pluripotency marker. At day

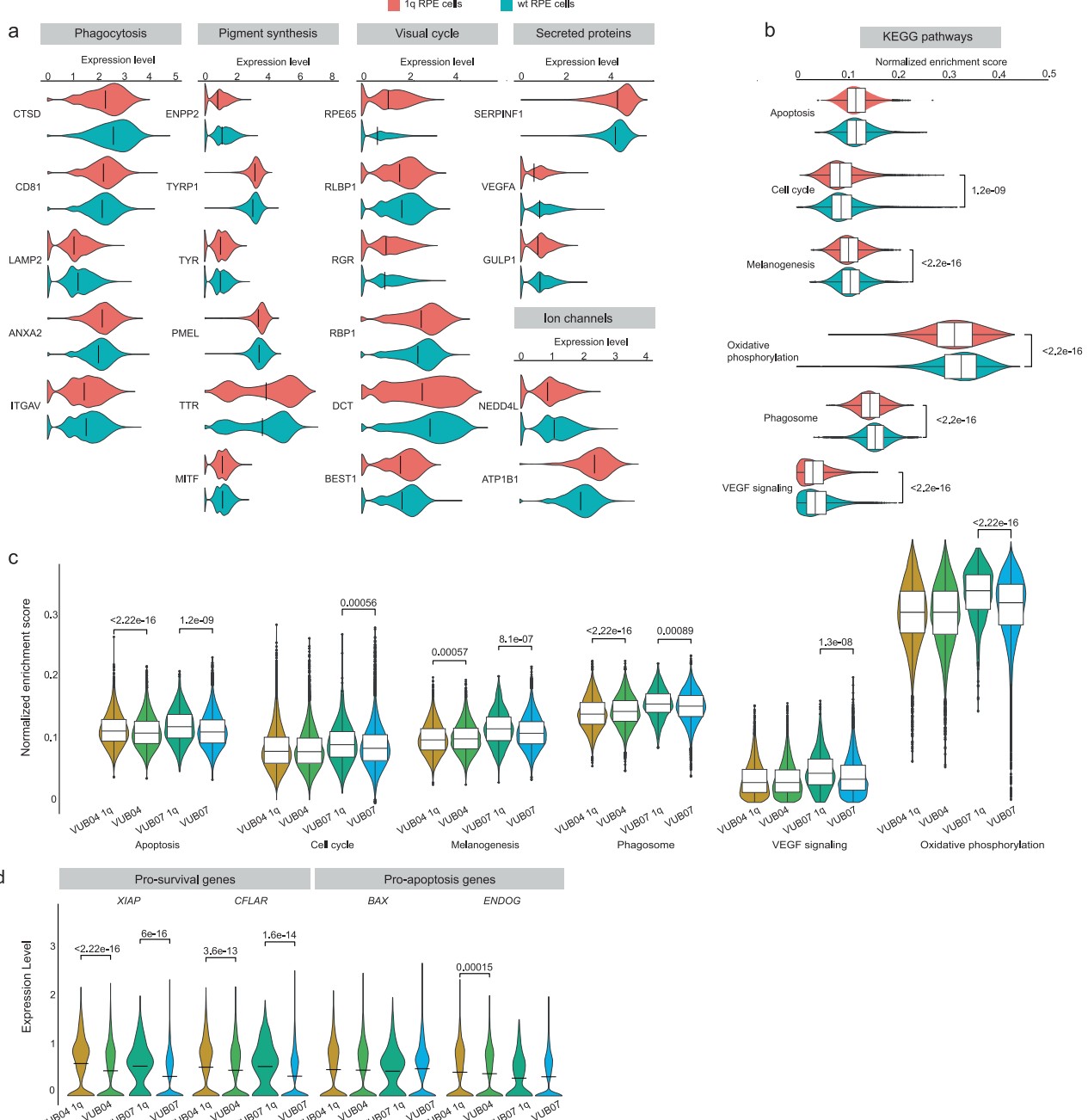

**Fig. 3 | Single-cell gene expression analysis of RPE with a gain of ¹q shows discrete transcriptional changes. a** Expression of genes characteristic of RPE¹q and RPEʷᵗ. All genes show statistically significant differences except for *MITF*. **b** scGSVA results for the KEGG libraries relevant to RPE cells. **c** scGSVA results for the libraries shown in (**b**), but comparing only the isogenic pairs of VUB04 and VUB07. **d** Expression of pro-survival, pro-apoptotic and effector genes, part of the apoptosis KEGG library, in the isogenic pairs of VUB04 and VUB07. In all panes, *P*-values shown above violin plots are unadjusted two-sided Wilcoxon rank-sum tests; for multi-gene comparisons, statistical significance was interpreted with Bonferroni correction. In **a** and **b**, n = 76.462 wt cells from 5 hPSC lines and n = 8732 cells with a 1q gain, from 2 hPSC lines. In **c** and **d**, numbers of cells are VUB04¹q = 8274, VUB04 = 10591, VUB07¹q = 458, VUB07 = 16658. The violin plots show a median line, and in **b** and **c**, 25th–75th percentile boxes.

14, only a small subset of neuroectodermal cells exhibited an RPE progenitor profile, marked by elevated expression of RPE-associated genes. These cells were more frequently observed in the VUB03ʷᵗ and VUB03¹q co-cultures compared to other conditions (for instance, 9% in VUB03¹q co-cultured vs 5% in VUB03¹q with iso20q or in monoculture). This is consistent with the previous findings, where prominent pigmented RPE cell clusters were observed only in the 2-month-old co-cultures of wt and 1q-gain cells, but not in 1q-gain monocultures or in any other aneuploidy tested.

## 1q RPE have a competitive advantage during differentiation but show aneuploidy-related stress

To confirm that gains of 1q confer a growth advantage during differentiation, we carried out co-culture experiments by introducing 0.2–0.5% of two fluorescently labeled hESC¹q lines alongside their genetically balanced isogenic counterparts (VUB03¹q32.1-Venus and VUB03¹q21.1qter-Blue, *N* = 6). The differentiation was followed by live fluorescent confocal imaging, which showed that 1q-gain cells rapidly outcompeted their genetically balanced counterparts (entire dish

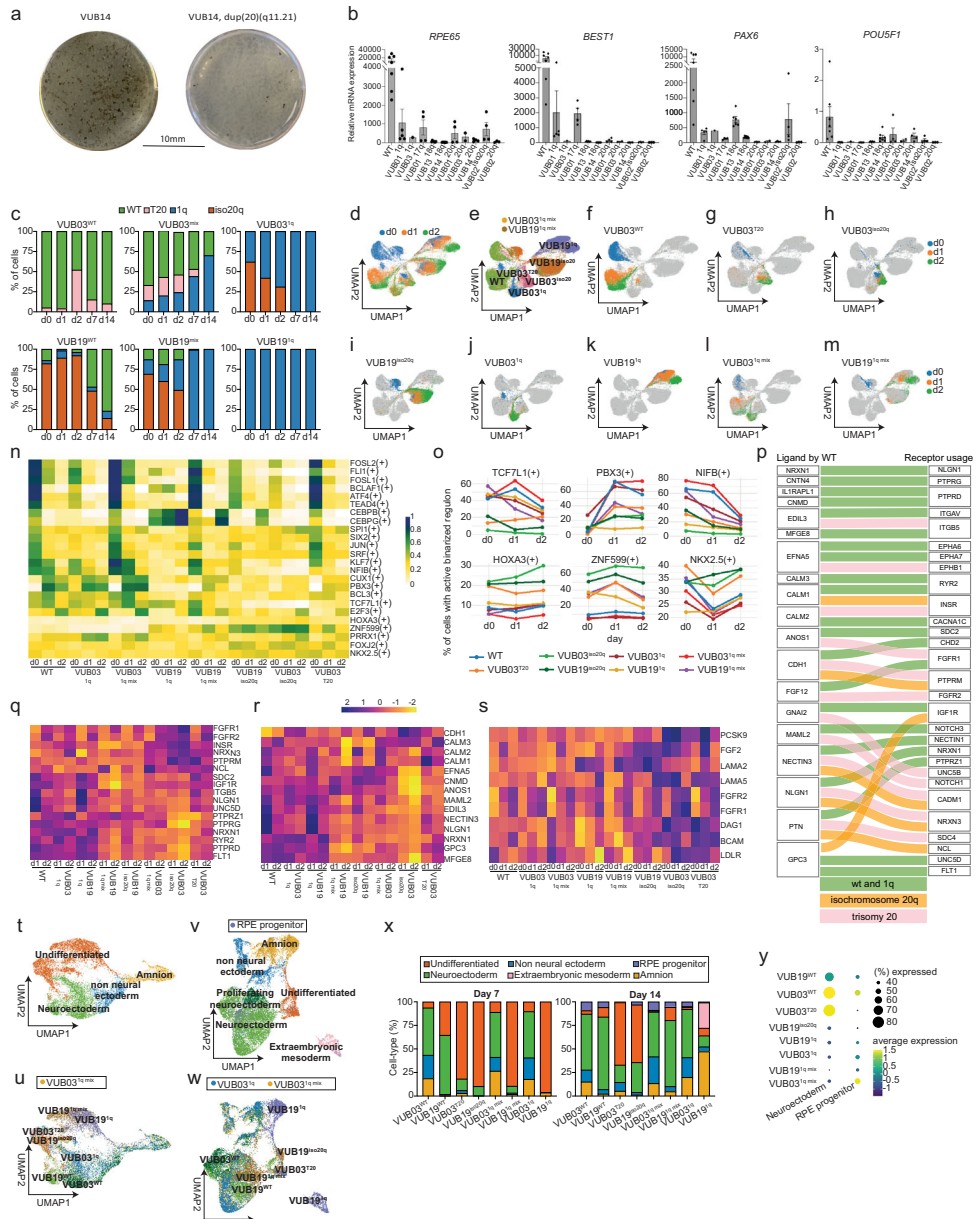

**Fig. 4 | Aneuploid lines fail to differentiate to RPE while co-culture with wt cells only changes the differentiation trajectory of [1q] cell. a** Representative image of pigmentation after 12 weeks of differentiation to RPE of control and aneuploid hESC. **b**. Expression levels of *RPE65*, *BEST1*, *PAX6* and *POU5F1* in cultures subjected to 12 weeks of RPE differentiation, prior to colony picking. The expression is relative to undifferentiated hESC. The error bars represent the mean expression ± SD of the different differentiation replicates, each indicated by a dot, *n* = 1–7 (Supplementary Table 6). WT indicates differentiation experiments of two different wt cell lines. **c** Bar plots showing the proportion of cells with different karyotypes across day 0 to day 14 (d0, d1, d2, d7, d14) of spontaneous differentiation. Iso20q refers to cells with an isochromosome 20q, T20 with trisomy 20, and 1q with a gain of 1q. **d–m** UMAP projections of the scRNAseq of differentiating cells from day 0 to day 2 of differentiation. **d** is colored by day of sampling and **e** by cell line identity. **g–m** The location per day for each karyotype and culture condition. **n** SCENIC top

regulon activity predictions across the first days of differentiation. **o** Line plots showing the percentage of cells positive for binarized regulon activity across karyotypes. **p** Sankey diagram representing ligand–receptor interactions. The ligands are secreted by wt cells and corresponding receptor usage in wt and 1q cells in co-culture (green), in iso20q cells (orange) and trisomy 20 cells (pink). Average expression of the receptors (**q**) and ligands (**r**) included in panel (**p**), filtered for those expressed in at least 70% of cells. **s** Average expression of ligands and receptors specific to the iso20q and trisomy 20 interaction networks. **t, u** UMAP projection of the scRNAseq at day 7 of differentiation, colored by cell type (**t**) and by cell line and karyotype (**u**). **v, w** UMAP projection of the scRNAseq at day 14 of differentiation, colored by cell type (**v**) and by cell line and karyotype (**w**). **x** Bar plots showing the distribution of cell types within each cell line and karyotype at day 14. **y** Gene set scores for neuroectoderm and RPE progenitor signatures within neuroectodermal subclusters, stratified by cell line.

images are shown in Fig. 5a). RPE cells were purified at 12 weeks, and RPE[1q] were isolated by flow cytometry. At the differentiation endpoint, the 1q-gain cell population had expanded 45- to 124-fold, comprising 9–24.9% of the total differentiated cells. These cells expressed key RPE markers ZO-1, PMEL, PAX6, and BEST1 similarly to control cells (Fig. 5b). We isolated RPE[1q] from the co-cultures by flow cytometry and

carried out bulk RNA sequencing. We included triplicates for each RPE[1q] line (VUB01[1q21.1q31.1-mKate], VUB03[1q32.1-Venus] and VUB03[1q21.1qter-Blue], *N* = 9) and ten control samples (5 lines, 2 replicates per line). Principal component analysis revealed that RPE[1q] samples clustered more closely (Fig. 5c), suggesting potential differences in their transcriptomic profiles. Differential gene expression analysis revealed that RPE[1q] cells

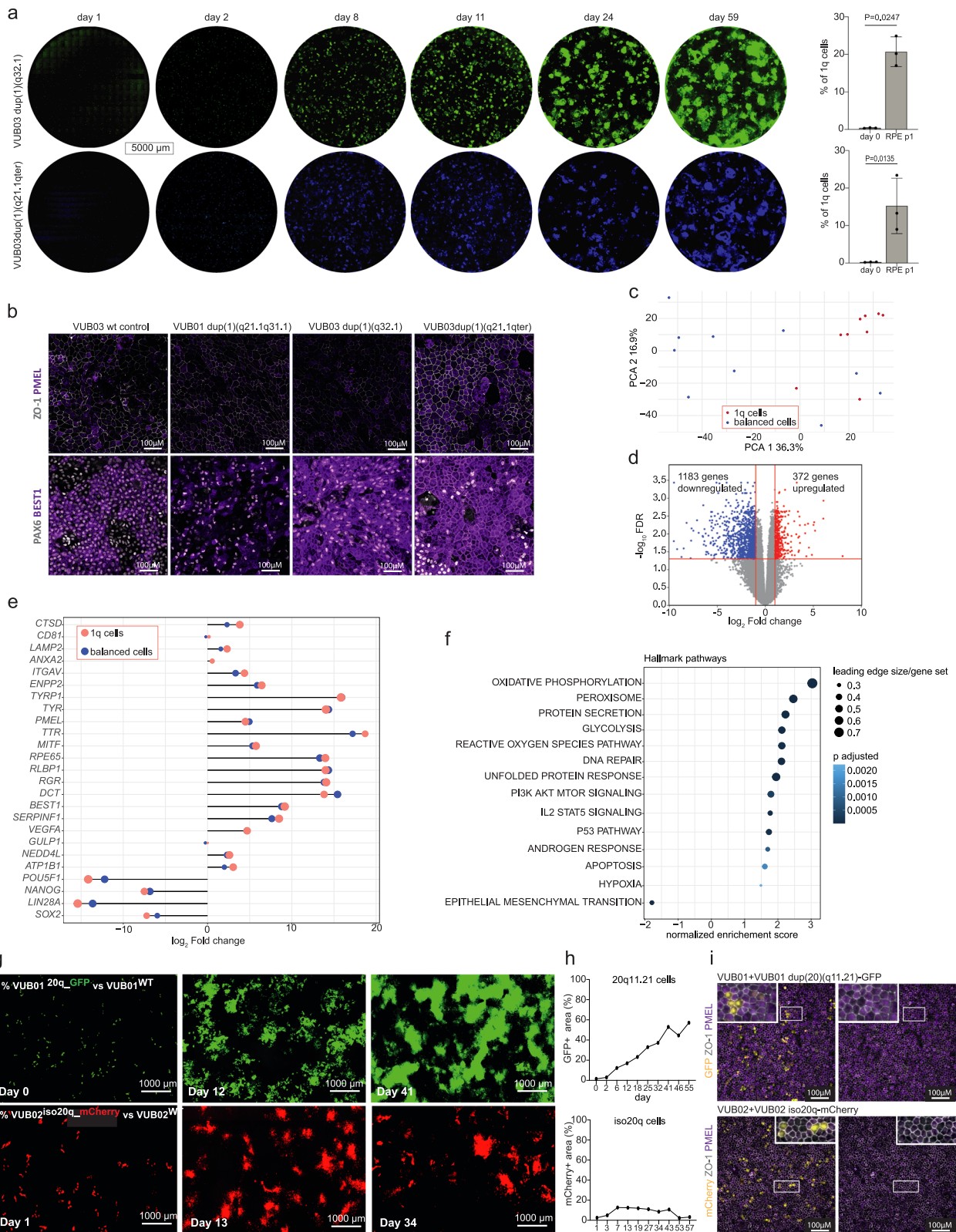

had upregulation of 372 genes and downregulation of 1183 genes compared to their RPE[wt] counterparts (Fig. 5d). We examined the induction patterns of RPE markers in both groups relative to undifferentiated hESCs and found that both control and 1q-gain cells showed similar upregulation of RPE differentiation genes alongside downregulation of pluripotency-associated genes (Fig. 5e). These findings align with the comparable RPE morphology and protein

expression patterns shown in Fig. 5b, confirming that hESC[1q] cells retain the capacity to differentiate into RPE cells similarly to genetically balanced counterparts, albeit only when in co-culture.

We further analyzed the differentially expressed genes by Gene set enrichment analysis and found that RPE[1q] cells showed deregulation of the apoptosis pathway, consistent with findings from the scRNA-seq data (Fig. 5f, Supplementary Data 1 and 2). Additional

**Fig. 5 | Cells with 1q and 20q11.21 gain outcompete wt cells during differentiation, but only RPE[1q] correctly specify despite showing transcriptomic signatures of aneuploidy-related stress. a** Images of the entire culture dishes during RPE differentiation of co-cultures of fluorescently labelled hESC with a gain of 1q and their isogenic genetically balanced counterparts (VUB03[1q21.1qter-BLUE] and VUB03[1q32.1-VENUS]). The plots indicate the percentages of 1q cells at the start and end of differentiation, mean and standard deviation, two passages after colony picking, as measured by flow cytometry (n = 3 different differentiation experiments, unpaired *t*-test). **b** Immunostaining for ZO-1, PMEL, PAX6, and BEST1 for the 1q cells isolated by flow cytometry (in **a**) and the RPE cells obtained from VUB03[1q32.21] (in

Supplementary Fig. 5). **c** Principal component analysis and **d** differential gene expression volcano plot for the bulk RNA sequencing of RPE[wt] (N = 10) and RPE[1q] (N = 9). **e** Fold-change induction of genes characteristic of RPE and of the pluripotent state, in RPE[wt] and RPE[1q], relative to undifferentiated hESC. **f** GSEA Hallmark pathways are significantly deregulated in the bulk RNA sequencing of RPE[1q] vs RPE[wt]. **g** Images of the culture dishes during RPE differentiation of co-cultures of 1% fluorescently labelled cells with a gain of 20q11.21 and an iso20q and their isogenic genetically balanced counterparts. **h** Percentage of GFP or mCherry-positive area over time in differentiation. **i** Immunostaining for ZO-1 and PMEL of the purified RPE cells, along with GFP and mCherry signal.

enriched pathways included the unfolded protein response, DNA repair, and p53 signaling pathways, processes commonly associated with intracellular stress responses triggered by aneuploidy[42]. These stress-related pathways likely contribute to the observed deregulation of apoptosis-related genes in both bulk and single-cell transcriptomic analyses. The only negatively enriched pathway was the epithelial-to-mesenchymal transition (EMT) gene set, an important process in RPE maturation[43].

## Co-culture does not rescue the differentiation of cells with 20q aneuploidy

To further confirm that genetically balanced cells cannot rescue the differentiation of aneuploid hESCs other than those with a 1q gain, we performed additional co-culture differentiation experiments. We mixed 1% of fluorescently labelled hESCs carrying either a 20q11.21 gain or an iso20q with their unlabeled, genetically balanced counterparts and initiated spontaneous differentiation. We imaged the differentiating cells up to the time-point of pigmented colony picking and quantified the area of fluorescent signal to estimate the percentage of 20q11.21 or iso20q cells (Fig. 5g, h). These results show that while the fraction of 20q11.21 cells increased steadily during spontaneous differentiation, the iso20q fraction initially increased, then stabilized and finally declined, in line with the declines we saw in the fractions of iso20q cells in the scRNAseq time courses.

We next purified the RPE population by manually picking pigmented colonies and expanding them to a homogeneous culture. The cells were stained for ZO-1 and PMEL, and co-imaged with GFP or mCherry to identify cells harboring the 20q11.21 gain or iso20q (Fig. 5i). We found that 3% and 5% of the resulting RPE populations consisted of cells with a gain of 20q11.21 or iso20q, respectively. In both cases, the aneuploid cells showed aberrant PMEL expression compared to their genetically balanced counterparts. Together, these results show that while the cells with a 20q11.21 gain can outcompete genetically balanced cells during spontaneous differentiation, the majority of correctly specified RPE originated from the genetically balanced cells. Iso20q cells, on the other hand, tend to be cleared out during differentiation, in line with our previous scRNAseq results during the first 14 days of differentiation (Fig. 4).

## Discussion

In this study, we show that spontaneous RPE differentiation is a purifying bottleneck against aneuploid cells, with the notable exception of cells with a gain of 1q. These cells can progress through differentiation when in co-culture with genetically balanced cells, which promote neural induction in the mutant cells.

In the first part of our work, we carried out single-cell high-resolution karyotyping of a cohort of 1674 individual hESC from whole genome sequencing-confirmed genetically balanced cultures. In comparison, in previous work, we used single-cell array-based comparative genomic hybridization (aCGH) to study 60 and 59 individual cells[13,14] and scDNAseq on a cohort of 56 cells[12]. This extended dataset refines the map of aberrations carried by hESC cultures in the form of low-grade mosaicism. We found that assumed genetically balanced

hESC cultures contain 3–6% of cells with chromosomal abnormalities–similar to our single-cell aCGH results[13], and lower than previously reported by us by scDNAseq[13] (23%) and by others by FISH[16] (1.3–2% of aneuploidy per chromosome tested). These differences are likely to reflect methodological differences in combination with variation across cell lines, different culture systems and variable passage numbers. In this dataset, we identified a broad range of genetic aberrations, including cells with a chaotic genetic content, much reminiscent of those found in cleavage-stage embryos[22], as well as aneuploidies that are recurrently found to take over hPSC cultures[2]. Remarkably, all three lines we screened carried cells with these recurrent aneuploidies, suggesting that all hESC cultures will eventually be taken over by any of these abnormalities, the speed of takeover depending on selective pressures from suboptimal culture conditions. We found aneuploidies in all chromosomes except for 19 and 22, which are also rare in hPSC cultures worldwide[2], suggesting these may be particularly deleterious.

When testing the effect of these abnormalities on RPE differentiation, we found that nearly all aneuploid hESC lines poorly specified to RPE, aligning with the published evidence of overall differentiation impairment of aneuploid hPSC, especially to neuroectoderm derivatives. For instance, gains of 1q and 20q11.21 impair directed neuroectoderm differentiation[6,44], cells with an iso20q are unable to differentiate to RPE[8] and losses of 18q impair anterior neuroectoderm differentiation[45,46]. Conversely, while only cells with a gain of 1q could correctly progress through spontaneous RPE differentiation when co-cultured with wt cells, the cells with a gain of 20q11.21, trisomy 20 or the iso20q enriched in alternative cell fates, remained undifferentiated or generated neuroectoderm-like cells with limited neural marker induction. It is important to note the technical constraints of this study and that undetected genetic variation, such as point mutations or very small CNVs, may also have influenced outcomes. Shallow whole-genome sequencing detects CNVs ≥0.5–1 Mb present in ≥20% of cells. Although we did not perform deep sequencing specifically for this work, in a separate study, we sequenced 380 cancer-related genes in most of the hESC lines used here. That analysis identified only a single low-frequency missense variant in an early-passage VUB03 culture, which was absent in later passages[47]. These findings, along with the reproducibility of the differentiation outcomes across many cell lines, support the conclusion that the observed effects are most likely driven by larger CNVs rather than smaller-scale genetic changes.

The time-course scRNA-seq experiments revealed that 1q-gain cells closely followed the wt differentiation trajectory when co-cultured, whereas trisomy 20 and iso20q cells diverged as early as the first 2 days after growth factor withdrawal. This early divergence led to significantly different cell populations by day 14, with RPE progenitor emergence and transcriptional maturity strongly influenced by these early cell fate decisions. These differences appear to be driven by the endogenous activation of distinct regulatory networks and differential expression of secreted ligands and receptors, such as FGF2 and FGFR2 in iso20q and trisomy 20 cells, as well as cell adhesion molecules including NLGN1, NRXN1, and NRXN3. Together, these findings suggest that only 1q-gain cells can be rescued in co-culture with wt cells

because they are transcriptionally close to wt cells and require only a subtle paracrine cue to initiate proper differentiation. In contrast, trisomy 20 and iso20q cells are intrinsically primed to maintain pluripotency and are unresponsive to paracrine cues from wt cells due to a lack of appropriate receptor expression. Overall, these findings align with the broader knowledge that RPE secretes trophic factors—including PEDF, bFGF, CNTF and VEGF—that promote retinal differentiation and homeostasis of receptive cells both in vivo and in vitro[48,49].

The results across this study consistently show that early spontaneous differentiation selectively favors 1q-gain cells over both wt and other aneuploid cells. This selective advantage is evident not only in terms of population-level expansion but also through the activation of key neuroectodermal regulons when co-cultured with wt cells, ultimately yielding RPE transcriptionally undistinguishable from wt, showing only mild signs of cellular stress when isolated. In contrast, other aneuploid cell types fail to initiate appropriate lineage-specific regulatory programs, and in some cases, still exhibit a selective advantage while contributing to mis-specified cell populations. Since the experiments in this study were not specifically designed to systematically assess the competitive advantage of each aneuploidy during spontaneous differentiation, our results pave the road for further research addressing this in a controlled and comparative experimental framework.

With regards to the mechanisms of selective advantage, gains of 20q11.21 gain advantage through overexpression of the antiapoptotic gene *BCL2L1*, located in the minimal region of gain[4,5]. It remains to be elucidated whether this mechanism is identical for the iso20q, but the results here suggest that it may not be the case, at least during spontaneous differentiation. Recent work by our lab and others has unraveled the mechanisms of the selective growth advantage of cells with a gain of 1q[44,50]. The common region of gain spans *MDM4*, a regulator of p53. Increased copy number of *MDM4* results in higher levels of MDM4 protein, leading to inhibition of the p53 response to DNA damage and lower downstream apoptosis. This selective advantage has been shown to be strongly influenced by culture conditions, and is retained during differentiation, with factors such as cell density and culture format having an important role in modulating competitive dynamics. Our findings are consistent with these reports, showing that 1q-gain cells maintain their selective advantage throughout long-term undirected differentiation toward RPE.

In this study, we used a spontaneous differentiation protocol. At the time the study began, this was the most used approach, including in clinical trials. However, spontaneous differentiation is time-consuming, labor-intensive, and inefficient for producing large quantities of RPE suitable for transplantation. More recently, directed differentiation protocols have been developed, which use exogenous factors or transcription factor expression to activate signaling pathways and guide cell fate more efficiently, reducing differentiation time[18,51]. Different protocols may alter the differentiation trajectory of hPSC, and consequently also influence aneuploid cell behavior, an important consideration when extrapolating our findings to other contexts.

Finally, while RPE with a gain of 1q obtained after co-culture appears morphologically and transcriptionally undistinguishable from their genetically balanced counterparts, this does not guarantee full functional equivalence. Further, when isolated and cultured alone, the cells show signatures of aneuploidy-related stress, such as p53 signaling, unfolded protein response and apoptosis. Transplantation experiments would be needed to evaluate their physiological performance, including their tumorigenic capacity, but these were beyond this study's scope.

In summary, RPE differentiation acts as a purifying bottleneck, removing most aneuploid lineages from hESC cultures except those with 1q gain. While rare aneuploidies are difficult to detect before

differentiation, 1q-gain cells expand and can be identified in the final product by genetic testing. Although RPE have low tumorigenic potential and no clinical trial has reported oncogenesis, the prevalence of mosaic aneuploidy in hPSC underscores the need for safety measures, such as targeted elimination of aneuploid cells[3] or synthetic kill switches[52]. Rigorous genomic surveillance and functional safeguards will be essential to ensure the genetic integrity and lineage fidelity of transplanted cells.

## Methods

### Resource availability
Further information and requests for resources should be directed to the corresponding author, Claudia Spits (claudia.spits@vub.be).

### Materials availability
All VUB stem cell lines in this study, including the genetically abnormal sublines and genetically modified lines, are available upon request and after signing a material transfer agreement.

### Ethics statement
For all parts of this study, the design and conduct complied with all relevant regulations regarding the use of human materials, and all were approved by the local ethical committee of the University Hospital UZ Brussel and the Vrije Universiteit Brussel (File number: B.U.N. 1432021000669). All hESC lines in this study were derived in-house in the past, and all patients donating embryos for the derivation of lines provided written informed consent.

### hESCs lines, cell culture, genomic characterization and banking
The details on the derivation and results of the characterization, including tests for pluripotency, were reported previously[53,54] and can also be found at the Open Science Framework repository [https://osf.io/esmz8/]. The lines are registered in the EU hPSC registry [https://hpscreg.eu/], and available upon request. Some of the lines were genetically modified to stably express a fluorescent protein, so that they could be traced in culture. This was achieved by lentiviral transduction of constructs encoding for mKate, GFP, mCherry, Pacific Blue or Venus. Briefly, Lentiviruses were produced in 293 T cells by transfecting plasmids for VSV.G, gag-pol and the plasmid of interest in PEI (1/28) (Sigma-Aldrich) and Opti-MEM (Thermo Fisher Scientific) for 4 h. The transfection cocktail was then replaced with complete medium, and the lentivirus-containing supernatant was harvested 48–72 h later and stored in aliquots at −80 °C. One day before transduction, hPSC were seeded at a density of 50,000 cells per well of a 6-well plate. Cells were then transduced in a transduction cocktail with 1:1000 protamine sulfate (LEO Pharma; 10 mg/mL) and a 50:50 mix of Nutristem and lentivirus-containing medium. In total, 800 μL of the cocktail was added to each well of a 6-well plate and incubated for 4 h. Cells were then washed with PBS 5× before adding fresh Nutristem medium. Twenty-four hours later, cells were again washed with PBS 5× and before being selected by either puromycin treatment, FACS or both.

Prior to the start of this study, cell working banks were created for each of the lines, which were karyotyped by shallow whole-genome sequencing and controlled for mycoplasma infection. The cells were cryopreserved in freezing medium composed of 90% knock-out serum (#10828, Thermo Fisher Scientific) and 10% dimethyl sulfoxide (DMSO, #D8418, Sigma-Aldrich) by slow cooling at −80 °C for 24 h, followed by storage in liquid nitrogen (LN$_2$). Cells were drawn from the bank for the experiments and used for differentiation either immediately after thawing, or not beyond 6 passages after thawing (Supplementary Table 1 and Supplementary Fig. 1).

The genetic content of the hESCs was assessed through shallow whole-genome sequencing by the BRIGHTcore of UZ Brussels, Belgium[55]. Cell lines were defined as "genetically balanced" if no CNVs

were detected, bearing in mind that this method has a resolution of 0.5–1 Mb and can detect mosaic CNVs at a level of ≥20%.

Copy number variant analysis for the gain of 20q11.21 was done using quantitative real-time PCR (qRT-PCR). DNA was extracted with a DNeasy Blood and Tissue Kit (Qiagen) according to the manufacturer's protocol. qPCR was performed with the copy number assays: *RNaseP* (Thermo Fisher Scientific) as a reference and *ID1* (Thermo Fisher Scientific) for the 20q regions. The reaction systems were prepared by mixing TaqMan 2× Mastermix Plus – Low ROX (Eurogentec) and the TaqMan assays together with the DNA samples. qPCR was performed on a ViiA7 thermocycler (Thermo Fisher Scientific), and Applied Biosystems Copy Caller v.2.1 was used to analyze the CNVs. The assay numbers are listed in the Supplementary Table 8. We previously published the outcomes of the targeted resequencing of 380 cancer-related genes to study a subset of our collection of lines and sublines. Four of the five genetically balanced lines used in this study were included in that analysis[47]. The results are included in Supplementary Table 1.

The hESCs were maintained in NutriStem hESC XF medium (NS medium; Biological Industries) with 100 U/mL penicillin/streptomycin (P/S) (Thermo Fisher Scientific) in a 37 °C incubator with 5% $CO_2$, on Biolaminin 521 coated dishes (Biolamina®). The culture medium was changed daily. The cells were passaged as single cells using TrypLE Express (Thermo Fisher Scientific) and split when reaching 70–90% confluence. The medium was supplemented with 10 µM Rho kinase (ROCK) inhibitor Y-27632 (ROCKi, Tocris) for the first 24 h after passaging.

## RPE differentiation
The cells were differentiated as described in ref. 16. HESC were cultured to confluence on Biolaminin-521®. Then, the cells were washed with PBS, and the medium was changed to NutriStem® hPSC XF GF-free with media changed every day. After 4 weeks, pigmented areas started to develop in the dishes. Following 8 weeks of differentiation, pigmented areas were mechanically cut out using a sharpened glass pipette. Then, cells were dissociated in accutase® for about 2 h. Cells were seeded on Biolaminin-521® coated dish and fed twice a week with NutriStem® hPSC XF GF-free. When passaged, the cells were dissociated using accutase® for 1h30-2h and then through a 30 µm strainer and seeded at a density of 50,000–100,000 cells/cm².

## Total RNA isolation, cDNA synthesis and quantitative real-time PCR (qRT-PCR) for gene expression analysis
Total RNA was isolated using RNeasy Mini and Micro kits (Qiagen) following the manufacturer's guidelines, including on-column DNase I treatment. mRNA was reverse-transcribed into biotinylated cDNA using the First-Strand cDNA Synthesis Kit (Cytiva) with the NotI-d(T)18 primer. Quantitative real-time PCR (qRT-PCR) was carried out using TaqMan mRNA expression assays (Thermo Fisher Scientific, listed in Supplementary Table 8) and TaqMan 2× Mastermix Plus – Low ROX (Eurogentec) on a ViiA 7 thermocycler (Thermo Fisher Scientific) using the standard settings provided by the manufacturer. The relative expression was determined by the comparative threshold cycle (Ct) method, and *GUSB* was used as the housekeeping gene. The assay numbers are listed in the Supplementary Table 8.

## Staining
RPE cells were fixed in 3.7% paraformaldehyde for 15 min, permeabilized in 0.1% Triton-X-100 for 10 min and blocked with 10% fetal bovine serum (FBS) for 1 h at room temperature. Between each step, the cells were washed 3× with PBS for 5 min. The primary antibodies were diluted in 10% FBS and kept at 4 °C overnight. The next day, and after 3× washing with PBS for 5 min, secondary antibodies were diluted 1:200 in 10% FBS for 1 h at room temperature. Nuclei were stained using a 1:1000 Hoechst dilution in PBS for 15 min. Finally, the cells were

washed 3× in PBS for 5 min. Confocal images were acquired under an LSM800 (Carl Zeiss) confocal microscope at 20× magnification. Supplementary Table 9 lists the antibodies used in this study.

## Live imaging and flow cytometry
For live imaging of mitoses in hESCs, 75,000 cells per well were seeded on glass-bottom µ-Plate Black 24-well plates (Ibidi) in Nutristem. Nuc-Blue™ Live ReadyProbes™ Reagent (Invitrogen, Thermo Fisher Scientific, 1 drop/10 mL) and SiR-Tubulin (Spirochrome, 100 nM) were added to the culture the next day to visualize DNA and microtubules. After 1 h incubation and without washing away the live imaging probes, the cells were imaged every 10–15 min for 15–18 h using a laser-scanning LSM800 confocal microscope (Zeiss) equipped with a fitted on-stage incubator in 5% $CO_2$ at 37 °C.

For flow cytometry, verified mycoplasma-negative cells were sorted under BSL-2 conditions using a 3-laser BD FACSMelody flow cytometer (BD Biosciences). Unlabeled cells served as a negative control to establish baseline fluorescence levels for gating to sort out BLUE or VENUS expressing cells. Sorting was performed based solely on BLUE/VENUS fluorescence, with singlets identified using FSC-W and SSC-W parameters. Analysis was done using Flowjo (BD Biosciences). An example of the gating strategy is shown in Supplementary Fig. 6.

## Single-cell DNA sequencing and CNV calling
hESC cultures were washed at least 3 times in PBS to remove all cell debris. The cells were dissociated using TrypLE™ for 15 min and strained through a 20 µm cell strainer and centrifuged for 5 min at 1000 rpm. Single-cell libraries were generated using the Chromium Single Cell CNV kit according to the manufacturer's instructions and sequenced on a Novaseq (Illumina) with a depth of 500k reads per cell. The fastq files were generated using cellranger-dna mkfastq (cellranger version 1.1.0) from the BCL file. The fastq were aligned to the Genome Reference Consortium Build 38 (GRCh38) using cellranger-dna cnv. The CNVs were determined using the R package Aneufinder (version 3.14). The results were kept only if the Bhattacharyya distance was ≥1 and the spikiness ≤0.20, and CNVs were only kept if ≥10 Mb. The heatmaps were generated using the R package copynumber (version 3.15).

## Single-cell RNA Sequencing and data analysis
The RPE cells were washed at least 3 times in PBS to remove cell debris and dissociated using Accutase for 1 h 30–2 h. The cells were strained through a 20 µm cell strainer and centrifuged for 5 min at 1000 rpm. Cells were resuspended in culture medium. Single-cell libraries were generated using the 10× Chromium Controller (10× Genomics) according to the manufacturer's instructions. Approximately 20k cells were sequenced per time point to minimize the occurrence of duplets. Sequencing was performed on a NovaSeq (Illumina) with 20k reads per cell (except one sample at 40k reads per cell). The fastq files alignment, filtering, barcode counting and UMI counting were performed using CellRanger 3.1.0 (10× Genomics). The reads were aligned to the Genome Reference Consortium Build 38. The different 10× genomics runs were aggregated using the CellRanger aggr pipeline. The processed scRNAseq were analyzed using (version 4.3) with the Seurat R package (version 4.3). Included cells in the analysis had nFeature count between 500 and 8000 and percent of mitochondrial genes <20%. The cell–cell variation was regressed based on the nCount_RNA, the percentage of mitochondrial gene content. The differential gene expression analysis was performed using the FindMarkers function from Seurat and the scGSVA package. Regulon analysis was carried out using Scenic[56], and the cell-to-cell communication analysis was done using Liana[57]. The inference of CNVs from scRNAseq was done using inferCNV[58]. The inferCNV run functions parameters were set to cutoff = 0.1, cluster_by_groups=T, HMM = T, analysis_mode was either set to «sub-clusters» or «cells» depending on the analysis, and denoise was set to *T*.

The other parameters were set to the default. For the analysis, we used an iterative manual approach to generate a reference set of genetically balanced cells. We generated Seurat-based clusters from the scRNAseq data and used one of them as a starting reference against the other clusters. We selected the clusters with no evidence of potential CNVs, used them against each other, and if still negative for CNV, added them to the reference set (a scheme of the workflow is shown in the Supplementary Fig. 7). This set was used against all other cells and allowed for the detection of the CNVs shown in this work.

## Statistics and reproducibility

All statistical tests were two-sided. In Fig. 1, all hESC lines were differentiated in multiple replicate experiments, as required to collect the necessary samples. The experiment shown in Fig. 1c was carried out once, on one cell line. The images and staining shown in Fig. 1f, g and Supplementary Fig. 2 were carried out once for each cell line. scRNA-seq shown in Fig. 1h–p was carried out once. scDNAseq was carried out for three lines. The imaging shown in Fig. 2f (and supplemental movie) was carried out once for one cell line. The experiments shown in Fig. 4a, b were carried out in multiple replicates per line; the details are listed in Supplementary Table 6. The experiments shown in Fig. 4c–y were carried out once, on six different culture conditions. The experiments shown in Fig. 5a were carried out three times per cell line, the staining in Fig. 5b were done once for each of the lines and the sequencing in Fig. 5c–f was done on three replicates for each 1q cell line (VUB01$^{1q21.1q31.1\text{-}mKate}$, VUB03$^{1q32.1\text{-}Venus}$ and VUB03$^{1q21.1qter\text{-}Blue}$) and two replicates of five control lines (VUB02, VUB03, VUB07, VUB14, and VUB32). The experiments shown in Fig. 5g–i were done for each cell line.

## Reporting summary

Further information on research design is available in the Nature Portfolio Reporting Summary linked to this article.

## Data availability

Raw sequencing data of human samples is considered personal data by the General Data Protection Regulation of the European Union (Regulation (EU) 2016/679), because SNPs can be extracted from the reads, and cannot be publicly shared. The data can be obtained from the corresponding author upon reasonable request and after signing a Data Use Agreement. The RNA sequencing count tables and all the data supporting the figures in this paper can be found at the Open Science Framework repository [https://osf.io/y8tzh/]. Source data are provided with this paper Source data are provided with this paper.

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

## Acknowledgements

Y.L. is a predoctoral fellow supported by the China Scholarship Council (CSC), and M.R., C.J., N.K., and E.C.D.D. are predoctoral fellows supported by the Fonds voor Wetenschappelijk Onderzoek Vlaanderen (FWO, grant numbers 1133622N, 11H9823N, 1169023N, and 1S73521N, respectively). M.C.D. is a predoctoral fellow supported by the 175 Military Hospital in Vietnam. This research was supported by the FWO (grant number G0713222N) and the Methusalem Grant to Karen Sermon and Claudia Spits (Vrije Universiteit Brussel).

## Author contributions

ECDD carried out all bioinformatic analysis and all wet-lab experiments unless stated differently. Y.L. carried out the RPE differentiation experiments of all aneuploid cell lines. N.K. generated the fluorescently labelled hESC lines and assisted with the cell culture work. D.A.D. assisted with the cell culture work and the writing of the paper. A.H. assisted with the cell culture work and the quantitative real-time PCR. M.C.D. assisted with the cell culture work during the revision of the paper. C.J. carried out the live imaging of hESC. M.R. assisted with the confocal microscopy imaging and figure preparation. S.V. carried out the flow cytometry quantifications. L.A.V.G. advised during the design and progress of the study and provided access to the flow cytometer. O.T. assisted with the genetic analysis of the cell cultures. K.M. supported the 10× Genomics Library preparation. K.S. provided funding for the study. C.S. designed, supervised and funded the study. All authors contributed to the drafting of the paper.

## Competing interests

The authors declare no competing interests.
