## [Transparent Peer Review file · Nature Communications]

1q gain bypasses the selective barrier against aneuploidy in RPE differentiation via wild-type co-culture rescue

Corresponding Author: Professor Claudia Spits

Version 0:

Reviewer comments:

Reviewer #1

(Remarks to the Author)

The manuscript by Deckersberg et al analyzed low level mosaicism of aneuploidy in ESC lines and showed that selectively 1q aneuploidy is enriched during spontaneous RPE differentiation. Authors go on to analyze if 1q aneuploidy in some cells is supported by mixed WT cells during differentiation and show that only 1q aneuploidy containing cells are supported – likely by some growth factors secreted by WT cells.

The manuscript analyzes a critical problem faced by all pluripotent stem cell-based therapies and raises the need to perform high content sequencing of such cell therapies to ensure patient safety. At the same time, the authors of this manuscript have the responsibility to provide sufficient data to ensure this manuscript doesn't raise false concerns for the field.

- 1) Authors use a spontaneous method of RPE differentiation, which differentiates less than 10% cells into RPE. What about the other 90% cells in the dish? Is aneuploidy a bottleneck only for RPE cells? What about other non-RPE cells? Is there any enrichment or not of any aneuploid state in those non-RPE cells?
- 2) It is not clear if aneuploidy mosaicism is a feature of ESC culture passage dependent, or if aneuploidy induction is a feature of this spontaneous differentiation process. Most ESCs used in this manuscript are late passage number. It is well-established that beyond passage 20 aneuploidy increases in ESC cultures. What if authors cultured ESCs for several passages without differentiation and tracked aneuploidy? In addition, what if they differentiated ESCs into a different cell type, would they see a similar degree of aneuploidy?
- 3) What is the mechanism of RPE-1q over-representation in RPE cells? Can authors identify potential cell cycle or anti-apoptosis genes and based on those genes test potential drugs/chemicals that would selectively kill such cells in early stages of differentiation. This would make these transplants safer and would be a productive outcome of this manuscript.
- 4) Is aneuploidy also induced in clinical-grade ESC and iPSC banks? All the ESC lines used in this manuscript are research grade. It is not clear if they are cultured under any standardized conditions akin to how it is done in clinical-grade settings. Authors should check widely used banks such as H1, H9 (both available from WiCell) and the Rutgers clinical-grade iPSC bank. This information will be valuable for the community.
- 5) Why are other aneuploidies not enriched during RPE differentiation? Again, is this specific to the RPE or to other cell types as well?
- 6) Authors claim that most variations between ESC lines are line-to-line variation. How can they be sure that variations seen in RPE-1q are due to 1q? How can authors rule out that the so called "line-to-line variation" is not due to other non-1q micro CNVs?
- 7) Significance of Figures 3e and f is not clear. These can be moved to supplemental.
- 8) The observation of NRXN1, 3, NLGN1, and MFGE8, and ECM proteins are not based on any validation experiment. Authors need to perform overexpression and KO experiments to prove that these factors do indeed regulate differentiation of RPE-1q cells.
- 9) Based on authors observation of aneuploidy related stress in 1q cells, it will be helpful for authors to provide some guidelines on how to identify such cells in culture. For instance, are RPE-1q functionally similar to native RPE cells and what percent 1q cells can be detected in an RPE culture and by what markers?
- 10) Are there any oncogenic genes on 1q chromosome arm that may provide selective advantage to form tumors? Authors need to perform tumorigenicity studies to determine if 1q cells indeed make tumors in an immunocompromised animal and what is the minimal percent mixture of 1q cells that will lead to tumors in transplants.

Reviewer #2

(Remarks to the Author)

The study by Couvreur de Deckersberg et al. is aimed at analyzing chromosomal aberrations in retinal pigment epithelium (RPE) differentiated from human pluripotent stem cells (PSCs). The authors differentiated several PSC lines that harbor a low percentage of aneuploidy at the undifferentiated state and suggest that during differentiation only cells with gain of 1q survive and take over the culture.

This is an interesting study that once comments are addressed, it may be suitable for publication.

Major comments:

1. The title hints that differentiation into RPE serves as a bottleneck against aneuploidies, while the paper really shows that cells with gain of 1q taking over the culture of differentiated RPE cells.
2. The authors mention that aberrations may “include point mutations and epigenetic changes”, but they only show chromosomal analysis. Since they have data on DNA analysis they should examine if their cell lines have point mutations that can induce selective advantage. Such mutations may influence their conclusions.
3. The authors refer to their control cells as “genetically balanced”. They should check their cells with a sensitive methodology for CNVs.
4. The paper states that out of the aneuploid cells only those with gain with 1q could differentiate into RPE. To validate it they state “we pick approximately 60 individual colonies per line, which appear to be predominantly clonal in origin”. Then they calculate that although aneuploidy appeared in low percentages, they should have detected them if they indeed differentiated into RPE. However, if the cells with 1q grow faster, the larger colonies, which are easier to pick up, will be of cells with 1q. Therefore, it is difficult to conclude that only (and not mostly) 1p cells differentiated into RPE. This should be stated in the Abstract, Results and Discussion.
5. Currently, all PSC-derived cells that are aneuploid should be avoided from transplantation. The authors invest a lot of effort to characterize the 1q RPE cells. Is the aim of the study to suggest that clinical grade cells with 1q should be allowed for transplantation? If yes, should we ignore the differences in transcriptome between RPE cells with and without 1q? If not, what is the value of the study for clinical trials?
6. The authors suggest that several genes may be involved in the growth advantage of 1q, but they haven't validated any of them. The paper will immensely benefit if the authors identified the gene/s in 1q that enable growth advantage.
7. The authors propose several genes that may be involved in cell-to-cell communication between wild type cells and 1q cells, but they haven't validated any of them. The paper will greatly benefit if the authors identified the gene/s involved in this process.

Minor corrections:

1. In the Introduction the authors state: “Over 50 clinical trials are ongoing or have recently been completed”, they probably refer only to clinical trials in the US.
2. The authors state in relation to RPE cells that “47% of these cells were in G2/M and 53% in S phase, indicating that they are proliferating cells”. Please cite other papers that state that all differentiated RPE cells are proliferating.
3. In the manuscript, the authors should refer to factors that can support RPE growth, e.g. PEDF (see Zhu et al. doi: 10.1167/iops.10-6413).
4. In Figure 1h, please give annotation to the large upper right (non-annotated) cluster.
5. Please correct the format of the references, see references number 2, 3, etc.

Reviewer #3

(Remarks to the Author)

Couvreur de Deckersberg et al. investigate the differentiation of human embryonic stem cells (hESCs) into retinal pigment epithelium (RPE) cells and identify it as a selective barrier against aneuploid cells. They show that differentiation into RPE eliminates most aneuploid cells, except those with specific genetic abnormalities, such as gains in chromosome 1q. Cells with a gain of 1q not only differentiate into RPE but also exhibit a growth advantage when co-cultured with genetically normal cells. RPE cells with a 1q gain (RPE1q) display subtle transcriptional changes, particularly in pathways related to

apoptosis and eye disease genes.

Overall, the paper is an interesting contribution to a comprehensive understanding of the genetic dynamics during RPE differentiation and highlights the role of co-culture in overcoming specific differentiation barriers.

However, I am concerned about how the authors perform the computational analysis of scRNA-seq CNV to identify the common 1q alterations. First, the main result is presented qualitatively in figures 2h-l, and the procedure presented in the methods section (lines 671-679) is not well described and is difficult to follow. I recommend rewriting this section using a more formal description in algorithmic terms, also publishing an R/python notebook to describe the main steps would be useful. Moreover, an orthogonal validation of these copy number calls using different methods such as CopyKat (PMID: PMID: 33462507) and/or SCEVAN (PMID: 36841879) would be useful.

Is there a 1q focal gain that is shared between the five cell lines?

They perform differential analyses between RPE1q and RPEwt, identifying 372 upregulated genes and 1183 downregulated genes. Are these genes enriched in 1q chromosomal locations? What if they exclude the genes in that location? Do they get the same functional enrichment?

How the authors identify the CNV in scDNAseq (figure 2a-c) is not described in the manuscript. They did not identify 1q gain the hESC lines. How do they exclude any sampling bias?

Version 1:

Reviewer comments:

Reviewer #1

(Remarks to the Author)

authors have addressed all of your reviewers comments.

Reviewer #2

(Remarks to the Author)

Reviewer #3

(Remarks to the Author)

The authors have responded satisfactorily to the criticisms.

Dear Reviewers,

Thank you for your constructive review and thoughtful feedback. We appreciate the time and care you have taken to evaluate our work, and your comments have been invaluable in strengthening this study.

In response, we have carried out substantial experimental and textual revisions. The most significant addition is a new, full differentiation dataset for an additional hESC line, profiled at days 0, 1, 2, 7, and 14 under three co-culture conditions, as well as expansion of the previous dataset to include day 7 and 14. This single-cell time-course now covers multiple karyotypes—wt, 1q gain, trisomy 20, and isochromosome 20q—across diverse co-culture contexts, enabling a more detailed and comparative view of differentiation trajectories. These new data directly address the concerns you raised and reveal novel insights into the mechanisms driving the divergent differentiation dynamics of these aneuploidies.

We have also revised the manuscript extensively to improve clarity and flow, restructured sections, and added two new figures (Figs. 4 and 5) containing substantial new data. The discussion now includes a more detailed consideration of the study's limitations and future directions. The most important textual changes are highlighted in red; other edits for conciseness were left unmarked to maintain readability while staying within the journal's 6,000-word limit.

We believe these additions and refinements have substantially strengthened the manuscript, and we are confident it now presents a complete and more compelling story. Below, we provide a detailed, point-by-point response to all comments.

Claudia Spits

REVIEWER #1

The manuscript by Deckersberg et al analyzed low level mosaicism of aneuploidy in ESC lines and showed that selectively 1q aneuploidy is enriched during spontaneous RPE differentiation. Authors go on to analyze if 1q aneuploidy in some cells is supported by mixed WT cells during differentiation and show that only 1q aneuploidy containing cells are supported – likely by some growth factors secreted by WT cells.

The manuscript analyzes a critical problem faced by all pluripotent stem cell-based therapies and raises the need to perform high content sequencing of such cell therapies to ensure patient safety. At the same time, the authors of this manuscript have the responsibility to provide sufficient data to ensure this manuscript doesn't raise false concerns for the field.

1) Authors use a spontaneous method of RPE differentiation, which differentiates less than 10% cells into RPE. What about the other 90% cells in the dish? Is aneuploidy a bottleneck only for RPE cells? What about other non-RPE cells? Is there any enrichment or not of any aneuploid state in those non-RPE cells?

We thank the reviewer for this important question. While our study focuses on RPE differentiation, we acknowledge that spontaneous differentiation yields a broad spectrum of cell types besides RPE. We performed single-cell RNA sequencing on two genetically balanced hESC lines following 60 days of spontaneous differentiation—the stage at which colonies are typically selected for RPE purification as part of a separate study but have now included these data in the current manuscript. This analysis, now included in Figure 1d, shows that in addition to RPE and retinal progenitor cells, the cultures also contain neural cell types such as cortical hem cells, oligodendrocyte progenitors, amacrine cells, and retinal ganglion cells, as well as a population resembling amnion-like cells (Figures 1d–e). These cell types are excluded from the final RPE cell product through manual colony picking and passaging.

Regarding the impact of aneuploidy on non-RPE cells, our data and that of others suggest that aneuploidy broadly impairs differentiation, not solely RPE specification¹. In the current study, we expanded our time-course experiments (now Figure 4) to include day 7 and day 14 samples, alongside days 0–2, and analysed additional karyotypic conditions. These new data support the conclusion that aneuploidy alters developmental trajectories in a karyotype-specific manner. For instance, isochromosome 20q and trisomy 20 appear to impair exit from pluripotency, whereas one of the 1q gain lines preferentially generates extraembryonic-like cells. Figure 5g-i further support this, showing that while 20q11.21 cells enrich during spontaneous differentiation, they do so in the non-RPE cells.

Overall, the results indicate that while aneuploidy broadly impacts differentiation potential in a karyotype-dependent manner, RPE are particularly sensitive to genetic imbalances, and that the specific dynamics of take-over depend on the aneuploidy and culture conditions. This has been further acknowledged in the manuscript, lines 383-391 and 414-423.

2) It is not clear if aneuploidy mosaicism a feature of ESC culture passage dependent, or if aneuploidy induction a feature of this spontaneous differentiation process. Most ESCs used in this manuscript are late passage number. It is well-established that beyond passage 20 aneuploidy increases in ESC cultures. What if authors cultured ESCs for several passages without differentiation and tracked aneuploidy? In addition, what if they differentiated ESCs into a different cell type, would they see a similar degree of aneuploidy?

Mosaicism is indeed a universal feature of cell culture, including stem cells. In our manuscript, we demonstrate this principle using single-cell DNA sequencing (scDNAseq) of hESC cultures that appear genetically balanced at the bulk level (Figure 2). As the reviewer noted, late-passage ESC cultures (beyond passage 20) are more prone to being overtaken by aneuploid cells with a growth advantage (e.g., 1q, 12p, 20q gains, and 18q deletions; see²). Conversely, the literature has not fully addressed whether mosaicism itself increases with passage, and it remains difficult to pinpoint the origin of the differences seen across studies.

We would like to clarify that the hESC lines used for Figures 1 and 2, as well as the RPE differentiation experiments, were primarily early to mid-passage lines: VUB02 (P8), VUB04 (P19), VUB07 (P27), VUB14 (P20), and VUB32 (P8) (as shown in Supplementary Table 1, Supplementary Figure 1). These lines were confirmed to be genetically balanced using shallow whole-genome sequencing, which detects copy number variations (CNVs) of 0.5–1 Mb at >20% mosaicism. The later-passage lines used in Figure 4 had been cultured over more passages to allow the aneuploidy to take over the whole culture, which in turn allowed us to assess their differentiation potential in the presence of specific aneuploidy. For instance, extended culture of VUB04 led to the takeover by the 1q cells initially identified via scDNAseq and later observed during RPE differentiation. The other lines were not cultured further, but these results align with the well-established tendency of aneuploid cells to dominate during extended culture.

With regards to differentiating to other cell types, we and others have extensively studied the impact of specific aneuploidies on differentiation. For example:

- Isochromosome 20q impairs RPE differentiation ³
- Gain of 20q11.21 impairs neuroectoderm differentiation but does not affect mesendoderm differentiation ^{4,5}.
- Gain of 12p disrupts hepatoblast differentiation by retaining residual pluripotency ⁶.
- Loss of 18q impairs neuroectoderm differentiation and delays cardiac progenitor induction ⁷.
- In work that was published in the time period this manuscript was under review, we show that gains of 1q retain their selective advantage during directed differentiation to the three lineages. While the aberration impairs differentiation to neuroectoderm (similarly as we see here), cardiac progenitors and hepatoblasts ⁸.

While we have not yet analysed the degrees of aneuploidy in differentiated products derived from genetically balanced cultures, the mosaicism observed in the undifferentiated state suggests similar outcomes to those presented here. We have further elaborated on this topic in the discussion lines 383-399 and 414-423

3) What is the mechanism of RPE-1q over-representation in RPE cells? Can authors identify potential cell cycle or anti-apoptosis genes and based on those genes test potential drugs/chemicals that would selectively kill such cells in early stages of differentiation. This would make these transplants safer and would be a productive outcome of this manuscript.

The mechanism of 1q take-over has been unravelled recently and published both by our group and that of Ivana Barbaric (Sheffield University)^{8,9}. Both studies show that the common 1q gain region includes *MDM4* – a regulator of p53 - which may promote cell survival by suppressing the p53-mediated DNA damage response and leading to reduced downstream apoptosis. However, although investigating potential strategies for selective elimination of such cells is important for future translational applications, therapeutic targeting of such cells is beyond the scope of the present study, which foremost focused on the impact of aneuploidy on differential potential.

We have included the references to the 1q work in lines 427-435, and discussed the idea of selective clearance of aneuploid cells by drugs (as proposed by for instance by Ben-David and Benvenisty¹⁰) or a kill-switch that would selectively eliminate transplanted cells displaying unwanted proliferative properties¹¹ in lines 453-455.

4) Is aneuploidy also induced in clinical-grade ESC and iPSC banks? All the ESC lines used in this manuscript are research grade. It is not clear if they are cultured under any standardized conditions akin to how it is done in clinical-grade settings. Authors should check widely used banks such as H1, H9 (both available from WiCell) and the Rutgers clinical-grade iPSC bank. This information will be valuable for the community.

Published data from the H1 and H9 lines (dataminid by Benvenisty's lab^{12,13}), and from WiCell itself shows pervasive aneuploidy in hPSC lines from labs worldwide⁹. In this last study, the authors report

on over 1500 lines and 23000 karyotypes, their results fully aligning with the known tendency of these cultures to become aneuploid.

Our lines are cultured in a fully defined culture system (Laminin-521 and Nutristem medium) and we work according to the ISSCR best practice standards for basic research <https://www.isscr.org/basic-research-standards>. All steps of cell culture are carried out based on SOPs, we have standardized moments for cell feeding and assessment of the cultures. Cells are counted during the cell culture splitting and split always at 70- 90% confluence (materials and methods, line 593-598).

Prior to the start of this study, cell working banks were created for each of the lines, which were karyotyped by shallow whole-genome sequencing and controlled for mycoplasma infection. Cells were drawn from the bank for the experiments and used for differentiation either immediately after thawing, or up to 6 passages after thawing (Supplementary Table 1 and Supplementary Figure 1).

5) Why are other aneuploidies not enriched during RPE differentiation? Again, is this specific to the RPE or to other cell types as well?

The results of our study support three conclusions. First, aneuploidy generally inhibits proper cellular specification to RPE. This aligns with other work by us and other groups, showing that this phenomenon is not restricted to RPE but is observed across various directed differentiation pathways^{3,4,7,8}. Our data also shows that aneuploidy-associated inhibition of RPE specification begins early in the differentiation process, where aneuploid cells fail to correctly initiate or sustain the molecular programs required for RPE lineage commitment. This is evident from the single-cell sequencing data in Figure 4, that shows the differentiation impairment of aneuploid cells other than 1q cells. Second, the cells with a gain of 1q appear to overcome this differentiation barrier when co-cultured with other cells. And third, our data shows that in co-culture, both cells with a gain of 1q and 20q11.21 exhibit selective growth advantages, dominating the culture over time (Figure 4). However, while 1q cells demonstrate the ability to correctly specify to the RPE lineage, the 20q11.21 cells enrich in the alternative cell populations.

Overall, there is a context-specific effect for each aneuploidy, and it is impossible to assert that all variants will take over under all conditions, and much research is required to map all possibilities. In the context of this study, it appears that at least gains of 1q and 20q11.21 will take over during spontaneous RPE differentiation, that all recurrent aneuploidies impair RPE differentiation, but that only the gains of 1q are able to bypass this impairment when co-cultured.

We now discuss this aspect in lines 383-399 and 414-423 of the manuscript and have added as a limitation of the study that we do not map out the ability for culture take-over for all the recurrent variants.

6) Authors claim that most variations between ESC lines are line-to-line variation. How can they be sure that variations seen in RPE-1q are due to 1q? How can authors rule out that the so called “line-to-line variation” is not due to other non-1q micro CNVs?

We thank the reviewer for raising this important point regarding the potential impact of undetected micro-CNVs on our findings and the differentiation between line-to-line variation and the effects of 1q gain.

We propose that the minimal gene expression differences observed between wild-type RPE cells are due to line-to-line genetic variation, as supported by previous studies¹⁴⁻¹⁶. While our karyotyping method detects CNVs down to 0.5–1 Mb with high confidence, we acknowledge the possibility that smaller undetected CNVs could contribute to this genetic variation. However, it is critical to note that

these differences occur across correctly specified cells (Figure 1), suggesting that any such variations do not significantly impact RPE specification.

The differences observed between wild-type and aneuploid cells, including those with 1q gain, are markedly larger than the minimal variations seen among wild-type lines. These differences manifest as severe mis-specification in aneuploid cells, a phenomenon consistently observed across the nine aneuploid lines we studied, and in the up to 7 replicate experiments per line. The magnitude and reproducibility of these differences, shows that 1q gain remains the main contributor of the variation.

Nevertheless, we acknowledge the limitations of our karyotyping approach in detecting CNVs smaller than 0.5–1 Mb, and the possibility that genetic variation other than the one detected modulates the outcomes of the differentiation. We have included this in the discussion (lines 393-399).

7) Significance of Figures 3e and f is not clear. These can be moved to supplemental.

We removed this part of the analysis.

8) The observation of NRXN1, 3, NLGN1, and MFGE8, and ECM proteins are not based on any validation experiment. Authors need to perform overexpression and KO experiments to prove that these factors do indeed regulate differentiation of RPE-1q cells.

We thank the reviewer for this important suggestion, which was also raised by reviewer 2. We agree that functional validation of candidate regulators such as NRXN1/3, NLGN1, and ECM components in the differentiation of RPE-1q cells would be of added value. In response, we undertook several experimental approaches to functionally test their relevance.

First, we attempted to modulate extracellular matrix composition by culturing the cells on various collagen combinations. Unfortunately, these conditions resulted in poor cell attachment, preventing further assessment of differentiation outcomes. We then screened for recombinant proteins corresponding to the identified candidate factors and identified a high-quality recombinant NLGN1 protein suitable for in vitro use. We exposed both wild-type and 1q gain hESC to NLGN1 during early spontaneous differentiation for one week. However, we observed no changes in anterior neuroectoderm marker expression. Wild-type cells continued to induce markers such as SIX3 as expected, whereas 1q cells failed to do so, and this phenotype was unaffected by NLGN1 treatment. These results suggest that NLGN1 alone is insufficient to restore normal differentiation trajectories in 1q cells under these conditions.

In parallel, we substantially expanded the scope of our study by including new single-cell RNA sequencing data across additional time points and karyotype contexts, and by performing refined analyses of regulon activity and cell–cell communication. These new data indicate that the ability of wild-type cells to rescue the differentiation of 1q cells in co-culture arises from a complex interplay of endogenous and exogenous factors. Notably, 1q cells can respond to an array of secreted and transmembrane proteins produced by the wt cells because they express the appropriate receptors. These are missing in other aneuploid cell types (e.g., iso20q or T20), which in turn also exhibit endogenously reinforced pluripotency networks, rendering them largely unresponsive to environmental modulation. Thus, while NLGN1 and the other candidate genes emerged as relevant from our computational analyses, it is likely that no single factor is sufficient to rescue the phenotype in isolation, making the functional validation very difficult and outside of the possibilities of the current study.

We have not included the negative results in the manuscript but hope that the new datasets and analyses and their discussion satisfactorily address this point.

9) Based on authors observation of aneuploidy related stress in 1q cells, it will be helpful for authors to provide some guideposts how to identify such cells in culture. For instance, are RPE-1q functionally similar to native RPE cells and what percent 1q cells can be detected in an RPE culture and by what markers?

We thank the reviewer for their insightful suggestion regarding the identification of 1q cells in culture and their functional similarity to native RPE cells.

Our data indicate that 1q cells, while in co-culture with wild-type cells, exhibit transcriptional profiles that are very similar to wt RPE and the aneuploidy related stress only becomes obvious in the bulk RNA sequencing of purified 1q RPE, after being isolated and grown on their own. We acknowledge that transcriptional similarity does not necessarily equate to functional equivalence. Functional validation, such as transplantation studies to evaluate their physiological performance, are warranted to address this question in greater detail. We have included this limitation in the discussion of the paper (lines 444-449).

Finding a specific transcriptomic signature in 1q cells when in co-culture with genetically normal cells could be a first step towards a method to identify 1q cells in culture. Unfortunately, we could not identify any specific markers for 1q cells that would allow their detection in heterogeneous cultures and can only suggest genomic methods such as FISH or digital droplet PCR, that have effective resolutions of 1-2%.

10) Are there any oncogenic genes on 1q chromosome arm that may provide selective advantage to form tumors? Authors need to perform tumorigenicity studies to determine if 1q cells indeed make tumors in an immunocompromised animal and what is the minimal percent mixture of 1q cells that will lead to tumors in transplants.

This is indeed an important question. As discussed in response to the reviewer's question 3, our recent work and that of Ivanna Barbaric's lab^{8,9} has shown that the selective advantage of 1q gain is mediated by increased expression of *MDM4*, a regulator of p53. The gain of *MDM4* suppresses the p53-mediated response to DNA damage, leading to reduced apoptosis and enhanced survival of 1q cells. This mechanism aligns with findings in cancer biology, such as Girish et al.¹⁷ which demonstrate that gain of 1q in cancers increases *MDM4* expression and suppresses p53 signalling, thereby promoting malignant growth.

While tumorigenicity studies in immunocompromised animals would provide valuable insights into the oncogenic potential of RPE with a gain of 1q, these assays fall outside the scope of our current work. Our study focuses on the impact of 1q aneuploidy on RPE differentiation and does not include transplantation experiments to evaluate tumor formation. It is also important to consider the following:

- There is a lack of residual undifferentiated cells in the purified co-cultured 1q cells, which reduces the likelihood of teratoma formation.
- RPE is inherently a low-tumorigenicity tissue, and thus transplanted RPE cells are unlikely to form tumors in a reasonable experimental timeframe. Even under extended observation, tumor formation might not occur without additional genetic or environmental factors.
- Detecting tumors driven by RPE-1q cells would likely require long-term studies involving a large cohort of animals. Such experiments, while undeniably valuable, are beyond the scope of our current investigation.

Future work could explore tumorigenicity in RPE-1q cells, potentially in collaboration with groups specializing in transplantation studies and cancer biology. These studies would address questions such as the minimal percentage of 1q cells required for tumor formation and their behavior in different tissue contexts. We have incorporated this in the revised manuscript, in lines 444-449.

REVIEWER #2

The study by Couvreur de Deckersberg et al. is aimed at analyzing chromosomal aberrations in retinal pigment epithelium (RPE) differentiated from human pluripotent stem cells (PSCs). The authors differentiated several PSC lines that harbor a low percentage of aneuploidy at the undifferentiated state and suggest that during differentiation only cells with gain of 1q survive and take over the culture.

This is an interesting study that once comments are addressed, it may be suitable for publication.

Major comments:

1. The title hints that differentiation into RPE serves as a bottleneck against aneuploidies, while the paper really shows that cells with gain of 1q taking over the culture of differentiated RPE cells.

The current title, "RPE Differentiation is a Selective Barrier Against Aneuploid hESC," reflects the broader observation that aneuploidy generally impairs RPE differentiation. However, as the reviewer points out, our findings also highlight the unique behavior of cells with a gain of 1q, which retain their selective growth advantage and overcome the differentiation bottleneck in RPE co-cultures.

To better capture these nuances, we could adjust the title to incorporate the idea of a different outcomes for the gain of 1q. The title for Nature Communications is limited to 15 words, so a possible alternative would be "1q gain bypasses the selective barrier against aneuploidy in RPE differentiation via wild-type co-culture rescue" This revised title highlights the general barrier function of differentiation while explicitly acknowledging the specific behavior of 1q cells in co-culture.

2. The authors mention that aberrations may "include point mutations and epigenetic changes", but they only show chromosomal analysis. Since they have data on DNA analysis they should examine if their cell lines have point mutations that can induce selective advantage. Such mutations may influence their conclusions.

The reviewer raises a very important point, into which we have already looked at the time of data analysis. The DNA sequencing methods we employed—shallow whole-genome DNA sequencing and scDNA sequencing—do not provide sufficient resolution to reliably detect point mutations. Specifically, the read depth for scDNA sequencing in the genetically balanced lines used was as follows: VUB07: 388 reads per Mb, VUB04: 756 reads per Mb, VUB02: 274 reads per Mb. For the shallow whole-genome DNA sequencing, this depth is in average 1000 reads per Mb. These read depths are insufficient to call point mutations.

We recently published an independent study¹⁸ that employed targeted resequencing of 380 cancer-related genes to study a subset of our collection of lines and sublines. Four of the five genetically balanced lines used in this study were included in that analysis. Among these, only VUB03 harboured a missense mutation in KMT2C with an allelic frequency of 0.05. Notably, later passages of VUB03 no longer carried this variant, suggesting it did not provide a strong growth advantage. Unfortunately, we do not have equivalent targeted sequencing data for all the aneuploid lines used in Figure 4, which examined differentiation impairment.

We have added the available sequencing data to Supplementary Table 1 and included a statement in the Discussion lines 393-399.

3. The authors refer to their control cells as "genetically balanced". They should check their cells with a sensitive methodology for CNVs.

As described in the Materials and Methods section, all hESC lines and sublines included in this study were analyzed using shallow whole-genome sequencing. This method has a resolution of 0.5–1 Mb and can reliably detect mosaic CNVs at a level of $\geq 20\%$. The karyotypes of all cell lines are detailed in Supplementary Table 1.

We have clarified in the Materials and Methods section that our definition of “genetically balanced” is based on the resolution of shallow sequencing lines 578-581. We have also added this to the discussion, lines 393-399.

We trust that this additional clarification ensures transparency regarding our methodology and its limitations.

4. The paper states that out of the aneuploid cells only those with gain with 1q could differentiate into RPE. To validate it they state “we pick approximately 60 individual colonies per line, which appear to be predominantly clonal in origin”. Then they calculate that although aneuploidy appeared in low percentages, they should have detected them if they indeed differentiated into RPE. However, if the cells with 1q grow faster, the larger colonies, which are easier to pick up, will be of cells with 1q. Therefore, it is difficult to conclude that only (and not mostly) 1p cells differentiated into RPE. This should be stated in the Abstract, Results and Discussion.

We thank the reviewer for pointing out the potential bias in colony selection due to the faster growth of 1q cells. We are happy to clarify our conclusions and hope that in the revised version of the manuscript this will be clearer.

In the scRNAseq experiments, we analyzed five RPE differentiation cultures, all starting from what appeared to be genetically balanced lines (based on shallow whole-genome sequencing). Three of these cultures resulted in fully genetically normal cell populations, with the caveat that inferCNV may not detect smaller CNVs. In the remaining two experiments, we observed a mixture of normal and 1q cells, with normal cells maintaining a substantial presence despite the growth advantage of 1q cells. This suggests that if other aneuploid populations had been capable of differentiating into RPE, they would likely have been detected, as genetically balanced cells were.

Additionally, our independent differentiation experiments further support our conclusion. Differentiation of 9 aneuploid hESC lines in a total of 37 experiments consistently resulted in poor RPE specification (Figure 4). When lines with isochromosome 20q or a gain of 20q11.21 were subjected to co-culture differentiation, only the 20q11.21 cells retained their growth advantage, but both consistently failed to differentiate into RPE. Instead, they remained in the mis-specified compartment of the dish. Figure 4 now includes these new data with mixed cultures, consistently showing that isochromosome 20q, trisomy 20 and monocultured 1q cells have alternative differentiation trajectories.

5. Currently, all PSC-derived cells that are aneuploid should be avoided from transplantation. The authors invest a lot of effort to characterize the 1q RPE cells. Is the aim of the study to suggest that clinical grade cells with 1q should be allowed for transplantation? If yes, should we ignore the differences in transcriptome between RPE cells with and without 1q? If not, what is the value of the study for clinical trials?

Thank you for the thoughtful comment. The goal of this part our study is not to suggest that clinical-grade cells with 1q gain should be allowed for transplantation. Rather, we aimed to explore whether there are detectable differences in transcriptome or other characteristics that might distinguish 1q cells from genetically balanced RPE cells. However, we did not find any transcriptional differences or markers that could distinguish these cells in a meaningful way.

Our findings suggest that 1q cells could potentially pass undetected in clinical settings if not carefully monitored and therefore be unwittingly transplanted, but we are not advocating for their use in transplantation. The current standard in clinical practice involves generating batches of cell products,

which are characterized based on predefined criteria to ensure they meet quality standards. Karyotyping is a routine part of this process and would likely detect abnormal cell populations such as those with 1q gain. The effectiveness of other phenotyping methods for detecting these anomalies remains uncertain.

6. The authors suggest that several genes may be involved in the growth advantage of 1q, but they haven't validated any of them. The paper will immensely benefit if the authors identified the gene/s in 1q that enable growth advantage.

This is indeed an important question, and, as discussed in response to Reviewer 1, our recent work and that of Ivanna Barbaric's lab^{8,9} has shown that the selective advantage of 1q gain is mediated by increased expression of *MDM4*, a regulator of p53. The gain of *MDM4* suppresses the p53-mediated response to DNA damage, leading to reduced apoptosis and enhanced survival of 1q cells. We also show that this mechanism of selective advantage is retained during differentiation, both directed and spontaneous differentiation, and in 2D and 3D systems.

We have added this to the discussion of the paper, lines 427-435.

7. The authors propose several genes that may be involved in cell-to-cell communication between wild type cells and 1q cells, but they haven't validated any of them. The paper will greatly benefit if the authors identified the gene/s involved in this process.

We thank the reviewer for this important suggestion, which was also raised by reviewer 1, and provide the same answer. We agree that functional validation of candidate regulators such as *NRXN1/3*, *NLGN1* and *ECM* components in the differentiation of RPE-1q cells would be of added value. In response, we undertook several experimental approaches to functionally test their relevance.

First, we attempted to modulate extracellular matrix composition by culturing the cells on various collagen combinations. Unfortunately, these conditions resulted in poor cell attachment, preventing further assessment of differentiation outcomes. We then screened for recombinant proteins corresponding to the identified candidate factors and identified a high-quality recombinant *NLGN1* protein suitable for in vitro use. We exposed both wild-type and 1q gain hESC to *NLGN1* during early spontaneous differentiation for one week. However, we observed no changes in anterior neuroectoderm marker expression. Wild-type cells continued to induce markers such as *SIX3* as expected, whereas 1q cells failed to do so, and this phenotype was unaffected by *NLGN1* treatment. These results suggest that *NLGN1* alone is insufficient to restore normal differentiation trajectories in 1q cells under these conditions.

In parallel, we substantially expanded the scope of our study by including new single-cell RNA sequencing data across additional time points and karyotype contexts, and by performing refined analyses of regulon activity and cell-cell communication. These new data indicate that the ability of wild-type cells to rescue the differentiation of 1q cells in co-culture arises from a complex interplay of endogenous and exogenous factors. Notably, 1q cells can respond to an array of secreted and transmembrane proteins produced by the wt cells because they express the appropriate receptors. These are missing in other aneuploid cell types (e.g., iso20q or T20), which in turn also exhibit endogenously reinforced pluripotency networks, rendering them largely unresponsive to environmental modulation. Thus, while *NLGN1* and the other candidate genes emerged as relevant from our computational analyses, it is likely that no single factor is sufficient to rescue the phenotype in isolation, making the functional validation very difficult and outside of the possibilities of the current study.

We have not included the negative results in the manuscript but hope that the new datasets and analyses and their discussion satisfactorily address this point.

Minor corrections:

1. In the Introduction the authors state: “Over 50 clinical trials are ongoing or have recently been completed”, they probably refer only to clinical trials in the US.
We have adjusted this to over a 100, and included the new update on clinical trials by Kirkeby, Main and Carpenter ¹⁹. The number of trials is different depending on which source we consult.
2. The authors state in relation to RPE cells that “47% of these cells were in G2/M and 53% in S phase, indicating that they are proliferating cells”. Please cite other papers that state that all differentiated RPE cells are proliferating.
We do not claim that RPE cells are proliferating, but rather that the cells in the very small cluster 2 are proliferating cells (Figure 1o). The vast majority of the cells in the scRNAseq dataset are in fact non-proliferating – cluster 2 corresponds to 1.32% of the cells.
3. In the manuscript, the authors should refer to factors that can support RPE growth, e.g. PEDF (see Zhu et al. doi: 10.1167/iovs.10-6413).
We have included the concept of factor secretion by RPE to the discussion, including this study and others that discuss these factors in vivo and in vitro. Lines 411-413.
4. In Figure 1h, please give annotation to the large upper right (non-annotated) cluster.
This cluster from the Senabouth data corresponds to residual undifferentiated cells (as evidenced by the presence of *POU5F1* and *LIN28A* positive cells) and to RPE cells of lesser maturation state (expressing lower levels of RPE markers such as *SERPINF1* and *MITF*).
5. Please correct the format of the references, see references number 2, 3, etc.
We have checked the references to the Nature Communications format.

REVIEWER #3

Couvreur de Deckersberg et al. investigate the differentiation of human embryonic stem cells (hESCs) into retinal pigment epithelium (RPE) cells and identify it as a selective barrier against aneuploid cells. They show that differentiation into RPE eliminates most aneuploid cells, except those with specific genetic abnormalities, such as gains in chromosome 1q. Cells with a gain of 1q not only differentiate into RPE but also exhibit a growth advantage when co-cultured with genetically normal cells. RPE cells with a 1q gain (RPE1q) display subtle transcriptional changes, particularly in pathways related to apoptosis and eye disease genes.

Overall, the paper is an interesting contribution to a comprehensive understanding of the genetic dynamics during RPE differentiation and highlights the role of co-culture in overcoming specific differentiation barriers.

However, I am concerned about how the authors perform the computational analysis of scRNA-seq CNV to identify the common 1q alterations.

- 1) First, the main result is presented qualitatively in figures 2h-l, and the procedure presented in the methods section (lines 671-679) is not well described and is difficult to follow. I recommend rewriting this section using a more formal description in algorithmic terms, also publishing an R/python notebook to describe the main steps would be useful.

We have provided a better description to the inferCNV approach, including the settings and the logic steps in the analysis process (lines 664-673 and supplementary figure 7), but cannot provide a notebook with the steps as much of the interpretation of the data was done manually. Developing an automated process to generate the reference set for the analysis and the further downstream curation of results would be indeed of value, but outside of the current scope of the study.

- 2) Moreover, an orthogonal validation of these copy number calls using different methods such as CopyKat (PMID: PMID: 33462507) and/or SCEVAN (PMID: 36841879) would be useful.

We agree that orthogonal validation of copy number calls is an important consideration. In addition to inferCNV, we tested both SCEVAN and CopyKat on our datasets. In our hands, inferCNV provided the most consistent detection of the 1q gain known to be present in the single cells used for the time-course experiments, as well as other aneuploidies of equal or larger size. For illustration, we show the raw inferCNV output for two undifferentiated VUB03 samples, where wild-type cells were compared to cells presumed to carry only the 1q gain. Under standard parameters, the corresponding SCEVAN heatmap did not detect the 1q gain or trisomy 20. While further optimization of SCEVAN settings might improve detection, we opted to proceed with inferCNV given its strong performance with default parameters in our dataset.

- 3) Is there a 1q focal gain that is shared between the five cell lines?

The hESC lines we used in this study are VUB01 dup(1)(q21.1q31.1), VUB03 dup(1)(q21.1qter), VUB03 dup(1)(q32.1), VUB19 dup(1)(q21.1qter). In the scRNAseq of the RPE we identified dup(1)q21.3q44 and dup(1)q21.2q24.2 in VUB04, and dup(1)q21.3q44 in VUB07. The scDNAseq revealed gain of 1q32.1q44 in VUB02 and of 1q21.3q44 in VUB04.

Now paying closer attention to the different variations in 1q gain, there appear to be two common regions in our data, something that has been already apparent from the data published by Baker et al in 2016. The common region for all of the gains of 1q, except for smallest gain in VUB03, spans q21.3 to q31.1. The distal breakpoint is here determined by the gain in VUB01. This region does not span the driver gene *MDM4*, which is conversely part of the small gain in VUB03. While all gains of 1q confer a competitive advantage to the cells, only the focal gain of VUB03 is not impairing the differentiation process. This highlights the q21.3q31.1 region as a promising area for further investigation into the gene(s) responsible for impaired differentiation.

- 4) They perform differential analyses between RPE1q and RPEwt, identifying 372 upregulated genes and 1183 downregulated genes. Are these genes enriched in 1q chromosomal locations? What if they exclude the genes in that location? Do they get the same functional enrichment?

We agree that this is an important question. Of the 1555 differentially expressed genes, 33 are upregulated and belong to the region of 1q gain. These genes are part of the leading edge of the predictions of 6 of the 14 enriched pathways, but in all cases are only one of the genes in the leading edge, representing less than 2% of the genes in the prediction each time. Overall, this suggests that the upregulated genes due to increased copy number are themselves not solely responsible for the transcriptomic profile but may rather act as upstream regulators that indirectly influence downstream gene expression.

How the authors identify the CNV in scDNAseq (figure 2a-c) is not described in the manuscript. They did not identify 1q gain in the hESC lines. How do they exclude any sampling bias?

The CNVs were detected using R package AneuploidyFinder (version 3.14). Cells were excluded if the results showed a Bhattacharyya distance smaller than 1 and a spikiness larger than 0.20. From the detected CNVs, we kept only those larger than 10Mb. This was mentioned in the materials and methods, but we have revised the methods section for clarity.

We found 1q gain in 9 single hESC, as shown in Figures 2a-c, the summary in Figure 1g and the supplementary table 5.

REFERENCES

1. Andrews, P. W. *et al.* The consequences of recurrent genetic and epigenetic variants in human pluripotent stem cells. *Cell Stem Cell* **29**, 1624–1636 (2022).
2. Amps, K. *et al.* Screening ethnically diverse human embryonic stem cells identifies a chromosome 20 minimal amplicon conferring growth advantage. *Nat Biotechnol* **29**, 1132–44 (2011).
3. Vitillo, L. *et al.* The isochromosome 20q abnormality of pluripotent cells interrupts germ layer differentiation. *Stem Cell Reports* (2023) doi:10.1016/J.STEMCR.2023.01.007.
4. Markouli, C. *et al.* Gain of 20q11.21 in Human Pluripotent Stem Cells Impairs TGF- β -Dependent Neuroectodermal Commitment. *Stem Cell Reports* **13**, 163–176 (2019).
5. Jo, H. Y. *et al.* Functional in vivo and in vitro effects of 20q11.21 genetic aberrations on hPSC differentiation. *Sci Rep* **10**, 1–15 (2020).
6. Keller, A. *et al.* Gains of 12p13.31 Delay WNT-Mediated Initiation of hPSC Differentiation and Promote Residual Pluripotency in a Cell Cycle Dependent Manner. *bioRxiv* <https://doi.org/10.1101/2021.05.22.445238> (2021) doi:<https://doi.org/10.1101/2021.05.22.445238>.
7. Lei, Y. *et al.* SALL3 mediates the loss of neuroectodermal differentiation potential in human embryonic stem cells with chromosome 18q loss. *Stem Cell Reports* **19**, 562–578 (2024).
8. Krivec, N. *et al.* Gain of 1q confers an MDM4-driven growth advantage to undifferentiated and differentiating hESC while altering their differentiation capacity. *Cell Death Dis* **15**, 852 (2024).
9. Stavish, D. *et al.* Feeder-free culture of human pluripotent stem cells drives MDM4-mediated gain of chromosome 1q. *Stem Cell Reports* **19**, (2024).
10. Ben-David, U. *et al.* Aneuploidy induces profound changes in gene expression, proliferation and tumorigenicity of human pluripotent stem cells. *Nat Commun* **5**, 4825 (2014).
11. Liang, Q. *et al.* Linking a cell-division gene and a suicide gene to define and improve cell therapy safety. *Nature* **563**, 701–704 (2018).
12. Lezmi, E., Jung, J. & Benvenisty, N. High prevalence of acquired cancer-related mutations in 146 human pluripotent stem cell lines and their differentiated derivatives. *Nat Biotechnol* (2024) doi:10.1038/s41587-023-02090-2.
13. Avior, Y., Eggan, K. & Benvenisty, N. Cancer-Related Mutations Identified in Primed and Naive Human Pluripotent Stem Cells. *Cell Stem Cell* **25**, 456–461 (2019).
14. Kilpinen, H. *et al.* Common genetic variation drives molecular heterogeneity in human iPSCs. *Nature* **546**, 370–375 (2017).
15. Abeyta, M. J. *et al.* Unique gene expression signatures of independently-derived human embryonic stem cell lines. *Hum Mol Genet* **13**, 601–608 (2004).

16. Ortmann, D. *et al.* Naive Pluripotent Stem Cells Exhibit Phenotypic Variability that Is Driven by Genetic Variation. *Cell Stem Cell* **27**, 470-481.e6 (2020).
17. Girish, V. *et al.* Oncogene-like addiction to aneuploidy in human cancers. *Science (1979)* **381**, 2023.01.09.523344 (2023).
18. Al Delbany, D. *et al.* De Novo Cancer Mutations Frequently Associate with Recurrent Chromosomal Abnormalities during Long-Term Human Pluripotent Stem Cell Culture. *Cells* **13**, (2024).
19. Kirkeby, A., Main, H. & Carpenter, M. Pluripotent stem-cell-derived therapies in clinical trial: A 2025 update. *Cell Stem Cell* **32**, 10–37 (2025).